# An arms race between 5'ppp-RNA virus and its alternative recognition receptor MDA5 in RIG-I-lost teleost fish

**Shang Geng[1], Xing Lv[1], Weiwei Zheng[1], Tianjun Xu[1,2,3]\***

[1]Laboratory of Fish Molecular Immunology, College of Fisheries and Life Science, Shanghai Ocean University, Shanghai, China; [2]Laboratory for Marine Biology and Biotechnology, Qingdao Marine Science and Technology Center, Qingdao, China; [3]Marine Biomedical Science and Technology Innovation Platform of Lin-gang Special Area, Shanghai, China

**Abstract** The incessant arms race between viruses and hosts has led to numerous evolutionary innovations that shape life's evolution. During this process, the interactions between viral receptors and viruses have garnered significant interest since viral receptors are cell surface proteins exploited by viruses to initiate infection. Our study sheds light on the arms race between the MDA5 receptor and 5'ppp-RNA virus in a lower vertebrate fish, *Miichthys miiuy*. Firstly, the frequent and independent loss events of RIG-I in vertebrates prompted us to search for alternative immune substitutes, with homology-dependent genetic compensation response (HDGCR) being the main pathway. Our further analysis suggested that MDA5 of *M. miiuy and Gallus gallus*, the homolog of RIG-I, can replace RIG-I in recognizing 5'ppp-RNA virus, which may lead to redundancy of RIG-I and loss from the species genome during evolution. Secondly, as an adversarial strategy, 5'ppp-RNA SCRV can utilize the m[6]A methylation mechanism to degrade MDA5 and weaken its antiviral immune ability, thus promoting its own replication and immune evasion. In summary, our study provides a snapshot into the interaction and coevolution between vertebrate and virus, offering valuable perspectives on the ecological and evolutionary factors that contribute to the diversity of the immune system.

## eLife assessment

This **important** study shows that in teleost fish, the RIG-I-like protein MDA5 can compensate for the absence of RIG-I by detecting 5'-triphosphorylated RNA. A fish virus containing such RNA can nevertheless evade MDA5 detection through a mechanism involving m6A methylation-induced silencing. The conclusions, which are supported by **solid** data, advance our understanding of antiviral immunity and virus-host conflicts in vertebrates.

## Introduction

The pattern recognition receptors (PRRs) recognize conserved pathogen-associated molecular pattern (PAMP) motifs, including proteins, lipids and nucleotides, resulting in activation of innate immune responses of host (*Pichlmair and Reis e Sousa, 2007*). Retinoic-acid-induced RIG-I-like receptors (RLRs) are a specific group of PRRs belonging to the family of DExD/H-box RNA helicases, and they play a crucial role in the innate antiviral response. There are three members in the RLR family: retinoic-acid-inducible gene I (RIG-I), melanoma differentiation-associated gene 5 (MDA5), and laboratory of genetics and physiology 2 (LGP2). RIG-I and MDA5 exhibit similar signaling characteristics and structural homology, including two N-terminal caspase-recruitment domains (CARDs), a DExD/H box

\*For correspondence:
tianjunxu@163.com

**Competing interest:** The authors declare that no competing interests exist.

**eLife digest** Before the immune system can eliminate a bacterium, virus or other type of pathogen, it needs to be able to recognize these foreign elements. To achieve this, cells in the immune system have proteins called pattern recognition receptors (PRRs) which can identify distinct molecular features of certain pathogens.

One specific group of PRRs is a family of retinoic acid-induced RIG-I-like receptors (RLRs), which help immune cells detect different types of viruses. Members of this family recognize distinct motifs on the genetic material of viruses known as RNA. For instance, RIG-I recognizes a marker known as 5'ppp on the end of single-stranded RNA molecules, whereas MDA5 recognizes long strands of double-stranded RNA.

Many vertebrates – including various mammals, birds, and fish – lost the RIG-I receptor over the course of evolution. However, Geng et al. predicted that some animals lacking the RIG-I receptor may still be able to activate an immune response against viruses that contain the 5'ppp-RNA motif.

To investigate this possibility, Geng et al. studied chickens and miiuy croakers (a type of ray-finned fish) which no longer have a RIG-I receptor. They found that both animals can still sense and eliminate two 5'ppp-RNA viruses called VSV and SCRV. Further experiments revealed that these two viruses are detected by a modified MDA5 receptor that had evolved to bind to 5'-ppp and activate the antiviral response.

Viruses are also continuously evolving new ways to escape the immune system. This led Geng et al. to investigate whether SCRV, which causes serious harm to marine fish, has evolved a way to evade the MDA5 protection mechanism. Using miiuy croakers as a model, they found that SCRV causes the transcripts that produce the MDA5 protein to contain more molecules of m6a. This molecular tag degrades the transcript, leading to lower levels of MDA5, reducing the antiviral response against SCRV.

The findings of Geng et al. offer valuable perspectives on how the immune system adapts over the course of evolution, and highlight the diversity of antiviral responses in vertebrates. Chickens and miiuy croakers are commonly farmed animals, and further work investigating how viruses invade these species could prevent illnesses from spreading and having a negative impact on the economy.

RNA helicase domain, and a C-terminal repressor doma in (RD) (*Yoneyama et al., 2005*). However, LGP2 lacks CARD domain (*Yoneyama et al., 2005*). The specific roles of RIG-I and MDA5 in response to virus stimulation are not redundant: RIG-I detects short blunt-end RNA with a 5'-triphosphate motif (5'ppp-RNA), while MDA5 specifically recognizes long forms of viral dsRNA (*Kato et al., 2006*; *Hornung et al., 2006*; *Loo and Gale, 2011*). In contrast, the involvement of LGP2 in cytosolic RNA sensing remains a topic of debate. Some studies have proposed that LGP2 is crucial for the production of type I interferon (IFN-I) in response to several RIG-I- and MDA5-dependent viruses (*Satoh et al., 2010*), whereas others have described LGP2 as a negative regulator of RIG-I signaling (*Saito et al., 2007*). However, surprisingly, despite the important immune functions of RLR members, they have been lost from the genome in many species including mammals, birds, and fish (*Xu et al., 2016*; *Liang and Su, 2021*; *Krchlíková et al., 2022*).

5'ppp-RNA is present in a wide variety of RNA viruses, including Flaviviridae, Paramyxoviridae, Coronaviridae, Orthomyxoviridae, and Rhabdoviridae families (*Hornung et al., 2006*). These viruses widely infect all kinds of life and cause various harmfulness and serious diseases, and the infected hosts range from the highest vertebrates to the lowest vertebrate, teleost fish (*Millet et al., 2021*; *Walker et al., 2022*). 5'ppp-RNA viruses are expected to be recognized and trigger an antiviral immune response by RIG-I. However, RIG-I is lacking in many vertebrates, so who then recognizes 5'ppp-RNA in these species? When a gene is functionally inactive or lost, other genes or pathways may undergo homology-dependent genetic compensation response (HDGCR) to maintain organismal balance. The pressure from viral infections has led to the evolution of the immune system, and the loss of RIG-I may be accompanied by functional substitution by homologous genes in order to compensate for the loss of RIG-I function. *Tupaia belangeri*, a mammal lacking RIG-I, has been reported to possess the MDA5 protein, which has undergone positive selection and exhibits the ability to sense Sendai virus (an RNA virus posed as a RIG-I agonist) for inducing type I IFN (*Xu et al., 2016*). This study provides

insights into the compensatory mechanism of RIG-I deficiency in vertebrates, however, considering the dynamic nature of host-virus dynamics, further research is urgently needed to validate and expand these insights.

For millions of years, viruses have coevolved with their hosts, leading to the development of various mechanisms for immune evasion in response to the mutual evolutionary pressure (*Naz et al., 2021*). Recently, it is reported that N6-methyladenosine (m⁶A) modification plays a crucial role in enabling viruses to evade the interferon (IFN) response (*Lou et al., 2021*). m⁶A methylation is an important epigenetic modification that involves the dynamic and reversible process of RNA methylation, regulated by methylase (writer), demethylase (eraser), and reader proteins that recognize m⁶A-modified RNA (*Shi et al., 2019*). N6-methylation deposition on mRNA primarily relies on the enzymatic 'writing' complex within the cell, with a key composition featuring methyltransferase-like 3 (METTL3) and METTL14. Conversely, N6-demethylation is chiefly modulated by 'eraser' proteins, with fat mass and obesity-associated protein (FTO) and AlkB homolog 5 (ALKBH5) being the identified enzymes for this role (*Zheng et al., 2013*; *Jia et al., 2011*). The functional impact of N6-methylation on RNA characteristics typically remains latent until recognition by 'reader' proteins, such as members of the YTHDF protein family, encompassing YTH domain family 1 (YTHDF1), YTHDF2, and YTHDF3. In mammals, YTHDF2 and YTHDF3 are recognized for their involvement in the degradation of m⁶A-modified RNA targets, as well as their association with mRNA localization and phase separation (*Wang et al., 2014*). By utilizing these m⁶A-related enzymes in the host, viruses can not only use m⁶A modification to hide themselves and avoid recognition by host receptors but also manipulate host gene expression to enable immune evasion (*Zhang et al., 2019*; *Lu et al., 2020*; *Wu et al., 2021*).

Fish and birds are commonly observed to exhibit 'RIG-I deficiency' phenomena, and they are also the most abundant groups of aquatic and terrestrial vertebrates respectively, giving them certain evolutionary advantages. As a result, the research of the immune evolution of them helps us to understand the immune evolution of the entire vertebrates. Meanwhile, fish and birds are major source of high-quality protein for human consumption, viral diseases in aquaculture and livestock industry pose serious obstacles to the industry's healthy development. Among them, the 5'ppp-RNA virus, as mentioned earlier, is a highly pathogenic group. Within this group, *siniperca cheats* rhabdovirus (SCRV) is an important pathogen that can cause huge economic losses and serious threats to aquaculture in freshwater and marine fish (*Zhang and Gui, 2015*). In addition, vesicular stomatitis virus (VSV) is also a highly infectious and widely prevalent 5'ppp-RNA virus (*Walker et al., 2022*). Therefore, in this study, we selected teleost fish miiuy croaker (*Miichthys miiuy*) and bird chicken (*Gallus gallus*), two vertebrates lacking RIG-I, as well as two 5'ppp-RNA viruses (SCRV and VSV), to investigate the evolutionary strategies and interactions between 5'ppp-RNA virus and host. The final results demonstrated that the MDA5 of *M. miiuy* and *G. gallus* lacking RIG-I both evolved the ability to recognize 5'ppp-RNA. In addition, SCRV virus can in turn utilize the m⁶A strategy to degrade MDA5 of *M. miiuy* for immune evasion. In summary, these findings shed light on the functional diversity of innate antiviral activity and unveil a complex arms race between virus replication and the innate immunity of its reservoir hosts in vertebrates. In addition, since the loss of RIG-I in vertebrates is a widespread event, our research also provides a new perspective for vertebrate groups that have lost RIG-I.

## Results
### A genetic compensation response model for RIG loss

In previous studies, it has been reported that a few vertebrates lost the RIG-I gene during evolution (*Liang and Su, 2021*; *Krchlíková et al., 2022*; *Chen et al., 2017*). To investigate this further, we analyzed the genomes of various species, and determined the presence and absence of the RIG-I gene. As depicted in *Figure 1A*, the absence of the RIG-I gene does not appear to be uncommon in vertebrates, with at least three independent loss events observed in mammals, birds and fish, respectively. These gene-loss events occurred over a long evolutionary timescale. Based on the minimum age of the common ancestor of RIG-I-missing species (http://www.timetree.org/), the oldest recorded losses occurring in species like *Erpetoichthys calabaricus*, can be traced back to approximately 330 million years ago (MYA). Conversely, the detected loss in *G. gallus* took place less than 42 MYA. To gain a deeper understanding of the evolutionary dynamics of vertebrate RIG-I, we explored the conservation of the genes surrounding RIG-I. We analyzed the genome sequences of *Homo sapiens*,

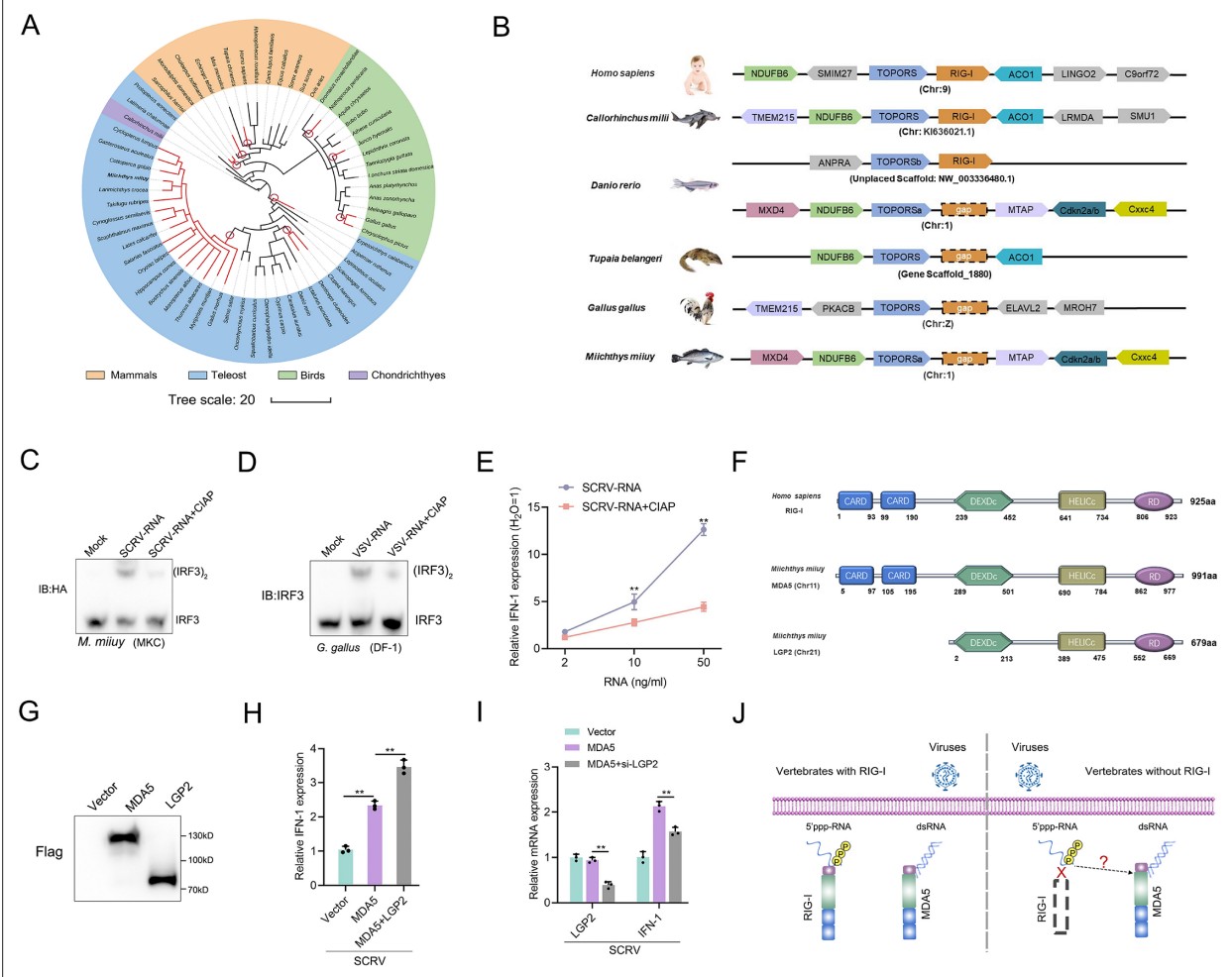

**Figure 1.** A genetic compensation response model for RIG loss. (**A**) Loss of RIG-I in vertebrates. Each branch tip represents one species. Red lines show lineages where loss of RIG-I. The red circle indicates the occurrence of an independent loss event. Branch lengths scale in millions of years (MYA). (**B**) Comparative analysis of gene synteny of RIG-I in vertebrate genomes. (**C and D**) SCRV-RNA and VSV-RNA were extracted from SCRV and VSV virus and dephosphorylated them with Calf Intestinal Alkaline Phosphatase (CIAP). Then, MKC cells were transfected with SCRV-RNA and SCRV-RNA +CIAP, and DF-1 cells were transfected with VSV-RNA and VSV-RNA +CIAP. IRF3 dimerization was analyzed by native gel electrophoresis. (**E**) MKC cells were transfected with different concentrations of SCRV-RNA and SCRV-RNA +CIAP, then the expression of IFN-1 was detected by qRT-PCR. As a control, the expression level of IFN-1 in MKC cells transfected with the same volume of water was set to '1'. (**F**) Predicted protein structures of MDA5 and LGP2 of *M. miiuy* and RIG-I of *H. sapiens*. (**G**) The expression of MDA5 and LGP2 plasmids of *M. miiuy* were detected by western Blot. (**H**) MKC cells were co-transfected with MDA5 plasmids and LGP2 plasmids for 24 hr and then treated with SCRV (MOI = 5) for 24 hr. The expression of IFN-1 was determined by qPCR. (**I**) MKC cells were co-transfected with MDA5 plasmids and si-LGP2 for 24 hr and then treated with SCRV (MOI = 5) for 24 hr. The expression of LGP2 and IFN-1 was determined by qPCR. (**J**) A genetic compensation response model for RIG loss. All data presented as the means ± SE from three independent triplicated experiments. \*\*, p<0.01, as determined by Student's t test.

The online version of this article includes the following source data for figure 1:

**Source data 1.** Excel files of data for *Figure 1*.

**Source data 2.** Raw unedited gels for *Figure 1*.

**Source data 3.** Uncropped and labeled gels for *Figure 1*.

*Callorhinchus milii*, *Danio rerio*, *T. belangeri*, *G. gallus*, and *M. miiuy* (*Figure 1B*). Similar to the slow-evolving *C. milii*, humans possess RIG-I with TOPORS and ACO1 as its upstream and downstream genes, respectively. As previously reported (*Xu et al., 2016*; *Krchlíková et al., 2022*). *T. belangeri*, *G. gallus*, and *M. miiuy* do not have RIG-I. Interestingly, zebrafish has a RIG-I with its upstream gene being TOPORSb, seemingly a result of chromosome fragment duplication. However, one RIG-I gene was lost in the downstream region of TOPORSa, indicating that one of the RIG-I copies has been lost in zebrafish, similar to *M. miiuy*.

Given the important role of RIG-I in recognizing 5'ppp RNA, we selected two different models for experiments to detect the presence of other potential receptors for immune compensation in RIG-I-deficient species. Specifically, we extracted virus RNA from SCRV and VSV viruses and obtained their dephosphorylated RNA derivatives (SCRV-RNA-CIAP and VSV-RNA-CIAP) through Calf Intestinal Alkaline Phosphatase (CIAP) treatment. Next, these RNAs were transfected into MKC cells of *M. miiuy* and DF-1 cells of *G. gallus* respectively to detect the dimerization form of IRF3. Results showed that virus RNA are more potent than dephosphorylated virus RNA in activating IRF3 (*Figure 1C and D*). And further qPCR experiments showed that the induction ability of IFN-1 by SCRV-RNA is significantly higher than that of SCRV-RNA-CIAP (*Figure 1E*). These results indicate the presence of viral receptors in *M. miiuy* and *G. gallus* that can recognize 5'ppp-RNA. Next, we compared the structural domains of human RIG-I and *M. miiuy* MDA5 (mmiMDA5) and LGP2 (mmiLGP2) proteins, and observed that mmiLGP2 lacks the CARD domain (*Figure 1F*). We next co-transfected mmiLGP2 plasmids or si-m-miLGP2 with MDA5 plasmids into MKC cells. The results showed that mmiLGP2 can actively regulate mmiMDA5's immune response to the SCRV virus (*Figure 1G–I*), which further indicates that it participates in antiviral immunity as an intermediate regulatory factor of MDA5 rather than a downstream signal transmitter (*Saito et al., 2007*). Therefore, these results prompted us to investigate whether MDA5, a structurally similar homolog to RIG-I, can substitute for RIG-I in recognizing 5'ppp-RNA in vertebrates without RIG-I (*Figure 1J*).

## MDA5 promotes host antiviral innate immunity

To further investigate the alternative immune function of MDA5, we first investigated the *M. miiuy* MDA5-mediated signaling pathway in response to SCRV virus and double-stranded mimetic poly(I:C)-HMW infection. Two siRNA molecules (si-MDA5-1 and si-MDA5-2) were designed to assess the function of MDA5. si-MDA5-1 demonstrated higher inhibitory efficiency compared to si-MDA5-2 (*Figure 2A*). Therefore, si-MDA5-1 (referred to as si-MDA5) was selected for subsequent experiments. Subsequently, we constructed the MDA5 expression plasmid and confirmed its protein overexpression efficiency and cytoplasmic localization (*Figure 2B and C*). Considering that mammalian MDA5 activates NF-κB and IRF3/IRF7 to induce IFNs, we investigated whether *M. miiuy* MDA5 could impact these pathways. Dual-luciferase reporter assays demonstrated that knockdown of MDA5 significantly inhibits NF-κB, IRF3, IRF7, and IFN-1 reporter genes (*Figure 2D*). In addition, we investigated the role of MDA5 in regulating the expression of IFN-stimulated genes (ISGs), which are crucial effectors. As shown in (*Figure 2E–G*), we observed that knockdown or overexpression of MDA5 can significantly inhibit or promote the expression levels of IFN-1, Mx1, ISG15, and Viperin upon SCRV infection or poly(I:C)-HMW stimulation. To visually represent the biological significance of MDA5 in SCRV-induced host cells, we utilized the virus plaque and qPCR methods to examine whether it could influence viral replication. As demonstrated in (*Figure 2H–K*), knockdown of MDA5 led to a significant increase in viral plaque formation and promoted SCRV replication in SCRV-infected cells, while overexpression had the opposite effect. Next, we explored that how MDA5 conducts signal transduction, and results showed that MDA5 is capable of binding not only to MAVS but also to STING (*Figure 2L and M*). In our previous laboratory investigations, we have substantiated the induction effect of STING on IFN under SCRV infection or poly(I:C) stimulation (*Chu et al., 2021*). Under the infection of SCRV virus, co-transfection of STING plasmids or si-STING with MDA5 plasmids can significantly increase or decrease the induction of IFN-1, indicating the important role of STING in MDA5-mediated immune transmission in response to RNA virus (*Figure 2N and O*). Finally, through the determination of IRF3 dimer and IFN induction, our results showed that the induction of antiviral innate immunity by SCRV-RNA depends on MDA5 (*Figure 2P and Q*). Overall, these findings indicated that MDA5 plays a positive role in regulating the antiviral responses of *M. miiuy*.

## RD domain is required for MDA5 to recognize SCRV RNA

We investigated whether *M. miiuy* MDA5, in the absence of RIG-I, can directly bind to the RNA of SCRV. To address this question, we precipitated the MDA5 protein from SCRV-infected MKC cells that were transfected with MDA5-Flag and pcDNA3.1-Flag. The bound RNA was then amplified using SCRV-specific primers or β-actin primers as a control. Our results showed that MDA5 interacted with SCRV RNA, while β-actin mRNA did not associate with MDA5 (*Figure 3A*). The C-terminal regulatory domain (RD) of human RIG-I binds to viral RNA in a triphosphate-dependent manner, to determine

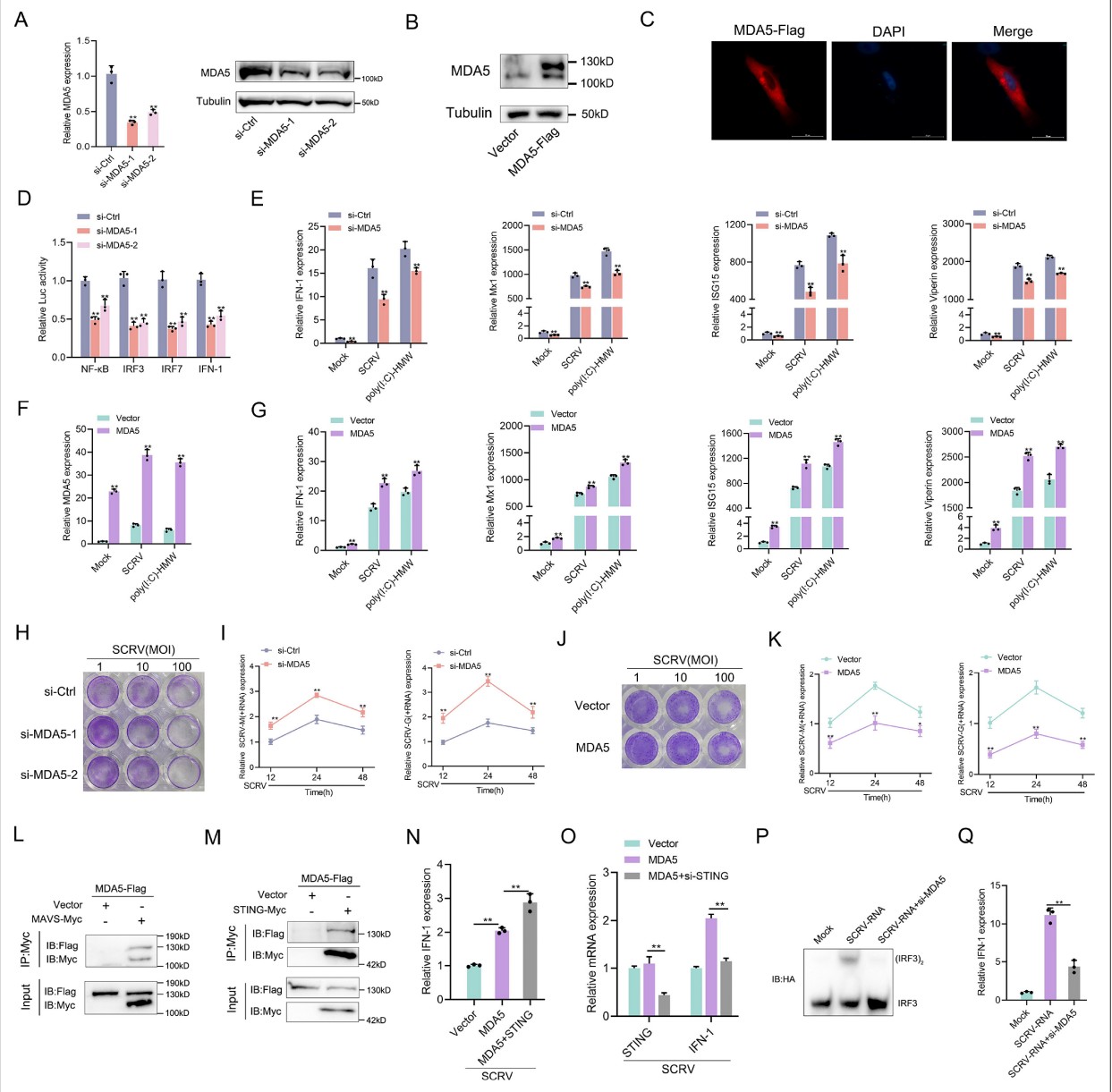

**Figure 2.** MDA5 promotes host antiviral innate immunity. (**A**) Silence efficiency of si-MDA5 measured by qRT-PCR and Western blotting. Two siRNAs of MDA5 were transfected into MPC cells for 48 hr, respectively. (**B**) MKC cells were transfected with MDA5 plasmids for 48 hr, then the overexpression efficiency was detected through MDA5 endogenous antibodies. (**C**) Subcellular localization of MDA5 in MKCs by immunofluorescence. (**D**) MDA5 knockdown suppresses NF-κB, IRF3, IRF7, and IFN-1 signaling. MPC cells were transfected with pRL-TK Renilla luciferase plasmid, luciferase reporter genes, together with MDA5 expression plasmid. At 48 hr post-transaction, the luciferase activity was measured and normalized to renilla luciferase activity. (**E**) Knockdown of MDA5 attenuates the expression of antiviral genes. MPC cells were transfected with si-Ctrl or si-MDA5 for 24 hr and then treated with SCRV (MOI = 5) or poly(I:C)-HMW for 24 hr. The expression of IFN-1, Mx1, ISG15, and Viperin were determined by qPCR. (**F and G**) Overexpression of MDA5 promotes the expression of antiviral genes. MKC cells were transfected with vector or MDA5 expression plasmids for 24 hr and then treated with SCRV (MOI = 5) or poly (**I:C**)-HMW for 24 hr. The expression of MDA5 (**F**), and antiviral genes, including IFN-1, Mx1, ISG15, and Viperin (**G**) were determined by qPCR. (**H and J**) MPC cells transfected with si-Ctrl or si-MDA5 and MKC cells transfected with pcDNA3.1 vector or MDA5 plasmid 24 hr and then treated with SCRV at the dose indicated for 48 hr. Then, cell monolayers were fixed with 4% paraformaldehyde and stained with 1% crystal violet. (**I and K**) MDA5 inhibits SCRV replication. MPC cells were transfected with si-Ctrl or si-MDA5 and MKC cells were transfected with pcDNA3.1 vector or MDA5 expression plasmid for 24 hr, then infected with SCRV (MOI = 5) for 12, 24, or 48 hr. The qPCR analysis was conducted for SCRV-M and SCRV-G RNA levels. (**L and M**) MDA5 immunoprecipitates with MAVS (**L**) and STING (**M**). EPC cells (~1 × 10$^7$) were co-transfected with MAVS-Myc or STING-Myc and MDA5-Flag expression plasmids for 24 hr, followed by immunoprecipitation (IP) with anti-Myc. (**N and O**) MKC cells were transfected with MDA5 plasmids and STING plasmids (**N**) or si-STING (**O**) for 24 hr and then treated with SCRV (MOI = 5) for 24 hr. The expression of IFN-1 was determined by qPCR. (**P and Q**) MKC cells were transfected with si-MDA5 for 24 hr and then transfected with SCRV-RNA

*Figure 2 continued on next page*

*Figure 2 continued*

for 24 hr, then the IRF3 dimerization and IFN-1 levels were analyzed by native gel electrophoresis (**P**) and qPCR (**Q**), respectively. All data presented as the means ± SE from three independent triplicated experiments. **, p<0.01; *, p<0.05, as determined by Student's t test.

The online version of this article includes the following source data for figure 2:

**Source data 1.** Excel files of data for *Figure 2*.

**Source data 2.** Raw unedited gels for *Figure 2*.

**Source data 3.** Uncropped and labeled gels for *Figure 2*.

if the RD domain of *M. miiuy* MDA5 has similar functions, we generated truncated mutations called MDA5-ΔRD and western blot analysis confirmed the expression of MDA5 and MDA5-ΔRD plasmids (*Figure 3B*). We subsequently noted that MDA5-ΔRD exhibited an inability to interact with SCRV RNA, which was supported by the loss of its immunological capability in inhibiting SCRV viral replication (*Figure 3C–E*). We also examined the effects of MDA5 and MDA5-ΔRD on antiviral factors in both uninfected, SCRV-infected, and poly(I:C)-HMW-stimulated MKC cells. In the absence of infection, overexpression of both MDA5 and MDA5-ΔRD stimulated the expression of antiviral genes (*Figure 3F–I*). However, when cells were infected or stimulated with SCRV or poly(I:C)-HMW, only the overexpression of MDA5, not MDA5-ΔRD, significantly increased the expression of antiviral genes (*Figure 3F–I*). In summary, these findings provided evidence that MDA5 may recognize SCRV RNA through its RD domain.

## MDA5 recognizes 5'ppp-RNA in *M. miiuy* and *G. gallus*

To provide specific evidence of MDA5 recognition of 5'ppp-RNA in *M. miiuy*, we synthesized 67 nt 5'ppp-SCRV from the SCRV virus and through in vitro transcription using T7 polymerase. As a positive control capable of binding to the mmiMDA5 receptor, 112 bp dsRNA from the MDA5 gene sequence was also generated. Gel analysis and nuclease sensitivity confirmed the generation of RNA products of the expected size (*Figure 4A*). The RNA pull-down experiment showed that 5'ppp-SCRV can be recognized by mmiMDA5, similar to 112 bp blunt end dsRNA (*Figure 4B*). Next, we transfected wild-type 5'ppp-SCRV and dephosphorylated mutant SCRV RNAs (5'-OH-SCRV and 5'pppGG-SCRV) into MKC cells to detect the dimer of IRF3, and found that the activation ability of 5'ppp-RNA on IRF3 depends on its triphosphate structure (*Figure 4C*). in addition, knockdown of mmiMDA5 blocked the induction of IRF3 by 5'ppp-SCRV (*Figure 4D*). Furthermore, biotin-labeled 5'ppp-RNA captured mmiMDA5, while the biotin-labeled mutant types (5'-OH-SCRV and 5'pppGG-SCRV) did not (*Figure 4E*). Additionally, we purified recombinant MDA5 and MDA5-ΔRD of *M. miiuy* to conduct electrophoresis mobility shift assay (EMSA) (*Figure 4F*). When recombinant mmiMDA5 was subjected to EMSA using a biotin-labeled 5'ppp-SCRV probe, a clear complex was detected (*Figure 4G*). This complex formation was inhibited by adding an excess of cold 5'ppp-SCRV but not 5'-OH-SCRV and 5'pppGG-SCRV, demonstrating the direct interaction between mmiMDA5 and 5'ppp-SCRV (*Figure 4G*). However, 5'ppp-SCRV cannot bind to the MDA5-ΔRD (*Figure 4H*). Next, we investigated whether *G. gallus* MDA5 (ggaMDA5) shares the ability to recognize 5'ppp-RNA seen in fish. We synthesized 67 nt 5'ppp-VSV from the VSV virus and found that it is similar to 5'ppp-SCRV, binding to ggaMDA5 and activating IRF3 in a triphosphate-dependent manner (4I-4M). Overall, our findings demonstrated that MDA5 detection of RNA in two RIG-I deficient vertebrates (*M. miiuy* and *G. gallus*) is guided by the RNA 5'-triphosphate end and is disrupted when RNA is capped at the 5'-triphosphate end or dephosphorylated.

## Altered m⁶A modification and expression of MDA5 upon SCRV infection

As stated in the previous results, MDA5 has evolved to possess immune recognition abilities that take over the role of RIG-I, significantly enhancing the host's ability to resist viral infections. However, evolution is a continuous battle between hosts and viruses, viruses often evolve complex escape mechanisms to evade immune surveillance and destruction, in which the m⁶A modification mechanism is an important way. To investigate the impact of SCRV infection on the overall methylation dynamics of fish hosts, we utilized methylated RNA immunoprecipitation sequencing (MeRIP-seq, GenBank accession number: PRJNA819945) to measure changes in m⁶A modification of host transcripts during

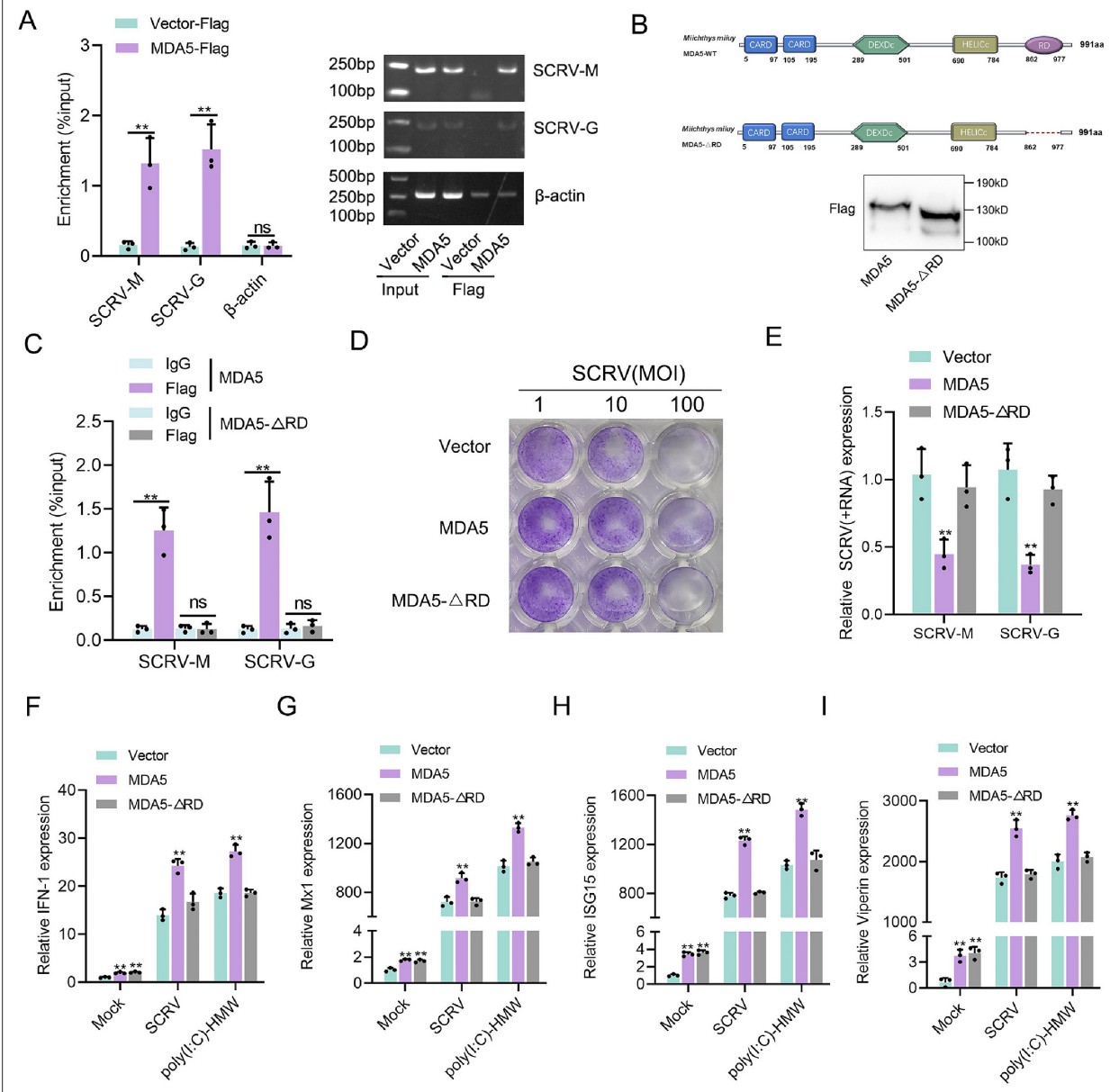

**Figure 3.** RD domain is required for MDA5 to recognize SCRV RNA. (**A**) The association of MDA5 proteins with SCRV. (**B**) Schematics and expression of MDA5 and truncated MDA5 (MDA5-△RD). (**C**) The association of MDA5 and MDA5-△RD proteins with SCRV. (**D**) MKC cells transfected with pcDNA3.1 vector, MDA5, or MDA5-△RD plasmids for 24 hr and then treated with SCRV at the dose indicated for 48 hr. Then, cell monolayers were fixed with 4% paraformaldehyde and stained with 1% crystal violet. (**E**) MKC cells were transfected with pcDNA3.1 vector, MDA5 or MDA5-△RD expression plasmids for 24 hr, then infected with SCRV for 24 hr, then the qPCR analysis was conducted for SCRV-M and SCRV-G RNA levels. (**F–I**) Expression of IFN-1 (**F**), Mx1 (**G**), ISG15 (**H**), and Viperin (**I**) in Mock-infected, SCRV-infected, and poly(I:C)-HMW-stimulated MKC cells transfected with pcDNA3.1 vector, MDA5, or MDA5-△RD expression plasmids. All data presented as the means ± SE from three independent triplicated experiments. **, p<0.01, as determined by Student's t test.

The online version of this article includes the following source data for figure 3:

**Source data 1.** Excel files of data for *Figure 3*.

**Source data 2.** Raw unedited gels for *Figure 3*.

**Source data 3.** Uncropped and labeled gels for *Figure 3*.

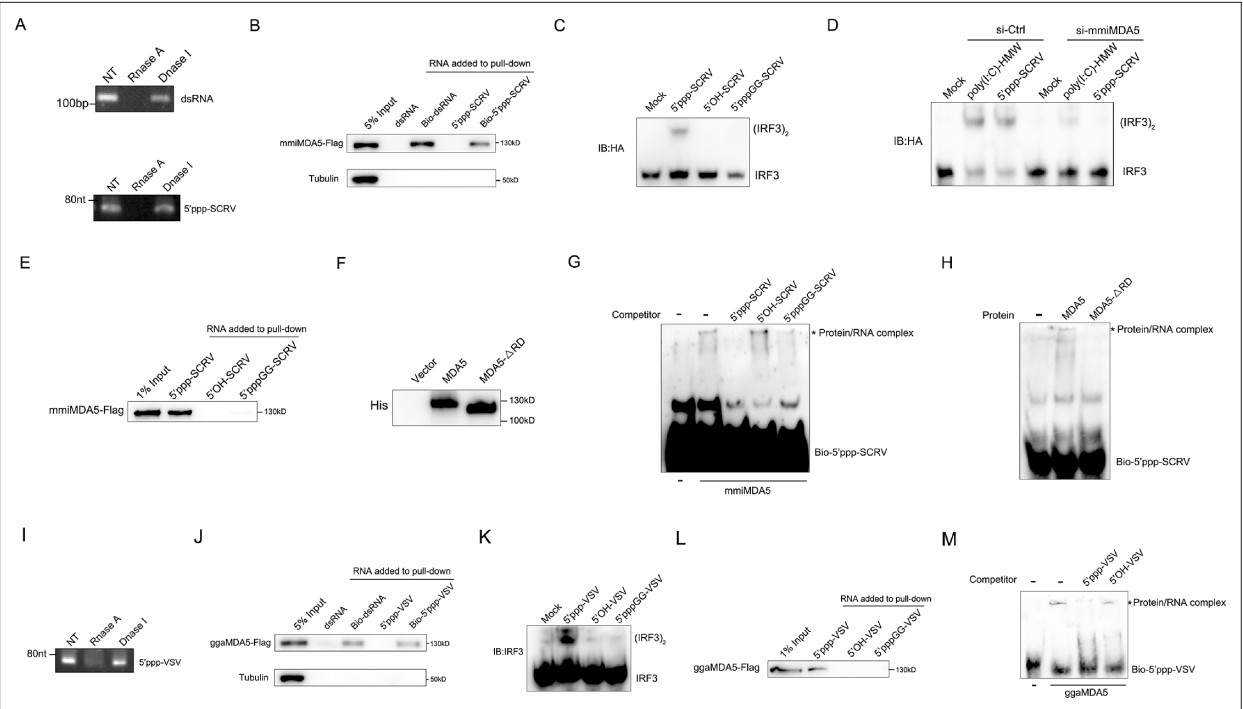

**Figure 4.** MDA5 recognizes 5'ppp-RNA in *M. miiuy* and *G. gallus*. (**A**) Schematic representation of 5'ppp-SCRV and dsRNA. The product of in vitro transcription runs as a single product degraded by RNase I. (**B**) Pull-down of biotinylated or non-biotinylated dsRNA and 5'ppp-SCRV by mmiMDA5. MKC cells were transfected with mmiMDA5-Flag plasmid, and the input and immunoprecipitated MDA5 proteins were analyzed by Western blot. (**C**) MKC cells were transfected with 5'ppp-SCRV, 5'OH-SCRV (5'ppp-SCRV dephosphorylated by CIAP), or 5'pppGG-SCRV (5'ppp-SCRV capped by m7G cap analog), then IRF3 dimerization was analyzed by native gel electrophoresis. (**D**) WT and si-MDA5 MKC cells were transfected with 5'ppp-SCRV and poly(I:C)-HMW, then IRF3 dimerization was analyzed by native gel electrophoresis. (**E**) The cytoplasmic fraction of MKC cells transfected with Flag-tagged mmiMDA5 was incubated with biotinylated 5'ppp-SCRV, 5'OH-SCRV, or 5'pppGG-SCRV. RNA-protein complexes were pulled down using streptavidin affinity beads. Input and pull-down samples were analyzed by SDS-PAGE and immunoblotting using anti-Flag antibody. (**F**) Purity of recombinant mmiMDA5 and mmiMDA5-ΔRD were determined by SDS-PAGE and immunoblotting using anti-His antibody. (**G**) EMSA of 5'ppp-SCRV with mmiMDA5. For binding competition, indicated non-biotinylated RNAs (50-fold molar excess over the probe) were included. The asterisk marks the binding bands between mmiMDA5 protein and different RNA probes. (**H**) EMSA of 5'ppp-SCRV with mmiMDA5 or mmiMDA5-ΔRD. The asterisk marks the binding bands between mmiMDA5 or mmiMDA5-ΔRD proteins and 5'ppp-SCRV RNA probes. (**I**) Schematic representation of 5'ppp-VSV. The product of in vitro transcription runs as a single product degraded by RNase I. (**J**) Pull-down of dsRNA and 5'ppp-VSV by ggaMDA5. DF-1 cells were transfected with ggaMDA5-Flag plasmids, and the input and immunoprecipitated MDA5 proteins were analyzed by Western blot. (**K**) DF-1 cells were transfected with 5'ppp-VSV, 5'OH-VSV (5'ppp-VSV dephosphorylated by CIAP), or 5'pppGG-VSV (5'ppp-VSV capped by m7G cap analog), then IRF3 dimerization was analyzed by native gel electrophoresis. (**L**) The cytoplasmic fraction of DF-1 cells transfected with Flag-tagged ggaMDA5 was incubated with biotinylated 5'ppp-VSA, 5'OH-VSV, or 5'pppGG-RNA. RNA-protein complexes were pulled down using streptavidin affinity beads. Input and pull-down samples were analyzed by SDS-PAGE and immunoblotting using anti-Flag antibody. (**M**) EMSA of 5'ppp-RNA with ggaMDA5. For binding competition, indicated non-biotinylated RNAs (50-fold molar excess over the probe) were included. The asterisk marks the binding bands between ggaMDA5 protein and different RNA probes.

The online version of this article includes the following source data for figure 4:

**Source data 1.** Raw unedited gels for *Figure 3*.

**Source data 2.** Uncropped and labeled gels for *Figure 3*.

SCRV infection (*Figure 5A*). From the results shown in *Figure 5B*, we identified 3060 differentially m6A methylated peaks (p-value<0.05) and 2082 differentially expressed genes (p-value<0.05) between the non-infected and SCRV-infected group, indicating that infection significantly alters the m6A modification of specific host transcripts. It is worth noting that hyper-methylated m6A peaks were also observed in the MDA5 mRNA transcripts during SCRV infection (fold change >1.5, p<0.05), and **t**he expression level of MDA5 was higher in the infection group compared to the control group (p<0.05). Further analysis of the m6A-seq data showed that MDA5 contained m6A sites in the exon1 region which were markedly higher in the infection group than that in the control group (Untreated VS SCRV: p=0.01; *Figure 5C*). Next, we tested the concrete m6A modification sites of MDA5 mRNA. We used

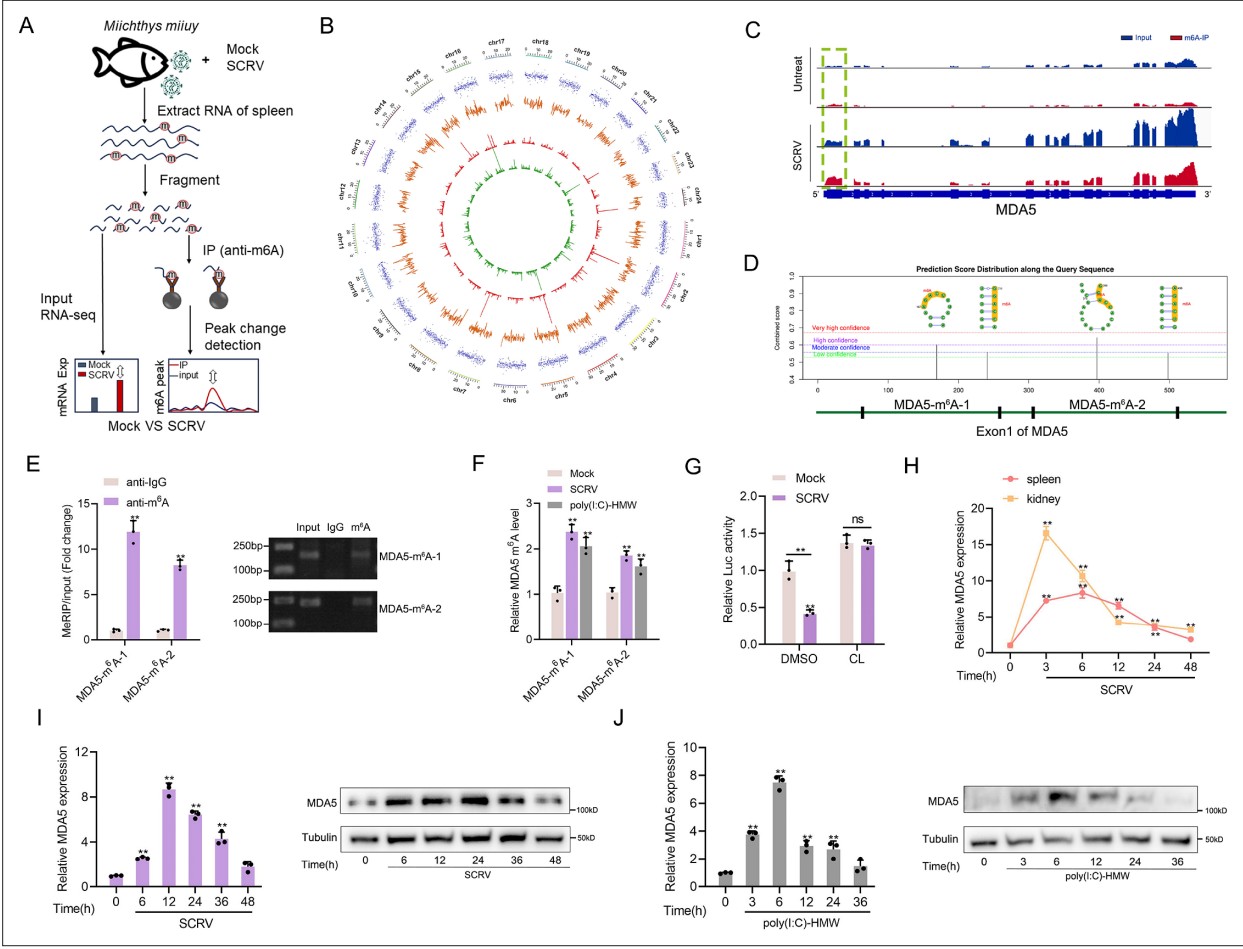

**Figure 5.** Increased m6A modification and expression of MDA5 upon SCRV infection. (**A**) Schematic of the MeRIP-seq protocol used to identify differential m6A methylation following infection of *M. miiuy* spleen tissues with SCRV. (**B**) Circos plots showing differentially m6A-methylated peaks identified from normal and SCRV-infected spleen tissues of *M. miiuy*. The green and red lines on the innermost ring represented the m6A peaks identified from normal and SCRV-infected spleen tissues of *M. miiuy,* respectively. The orange lines represent the log2 fold change values for each differentially m6A peak. The blue spots represent the fold change of gene expression. Chromosomes were shown on the outermost ring, the ruler on which represented the physical distance is in millions of bases (Mb). (**C**) The m6A abundance in MDA5 mRNA transcripts detected by MeRIP-seq, the m6A peak of MDA5 is circled in the green box. (**D**) The exon1 sequence of MDA5 was submitted to the SRAMP website, and then the predicted m6A site was displayed. Four predicted methylation sites located on the exon1 of MDA5 were marked by red box, then we designed two m6A specific primers (MDA5-m6A-1 and MDA5-m6A-2). (**E**) m6A abundance on MDA5 exon1 detected by MeRIP-qPCR and emiquantitative PCR in MKC cells. (**F**) MeRIP-qPCR analysis of relative m6A level of MDA5 in Mock, SCRV-infected, and poly(I:C)-HMW-stimulated MKC cells. (**G**) Relative luciferase activity of pmirGLO-MDA5-exon1 firefly luciferase reporters in Mock and SCRV-infected (MOI = 5) MKC cells that treated with DMSO or Cycloleucine (CL, 20 mM). (**H**) mRNA levels of MDA5 in spleen and kidney tissues measured by qRT-PCR at indicated time after SCRV infection (MOI = 5). (**I and J**) mRNA and protein levels of MDA5 in MKC cells measured by qRT-PCR and Western blotting at indicated time after SCRV infection (MOI = 5) (**I**) and poly(I:C)-HMW stimulation (**J**). All data presented as the means ± SE from three independent triplicated experiments. **, p<0.01, as determined by Student's t test (E-G) and one-way ANOVA with Dunnett's multiple comparisons test (H-J).

The online version of this article includes the following source data for figure 5:

**Source data 1.** Excel files of data for *Figure 5*.

**Source data 2.** Raw unedited gels for *Figure 5*.

**Source data 3.** Uncropped and labeled gels for *Figure 5*.

the SRAMP database to predict potential m6A sites in MDA5-exon1 (*Zhou et al., 2016*). Four potential sites were predicted and two specific primers were designed (*Figure 5D*), then m6A quantitative real-time PCR and semiquantitative PCR were performed. In line with the m6A-seq data, the results further confirmed that m6A sites exactly exist in MDA5 exon1 (*Figure 5E*). After being infected with SCRV or poly(I:C)-HMW, the methylation level of MDA5 increased in MKC cells, further confirming the

sequencing data (*Figure 5F*). We further inserted the MDA5-exon1 into pmirGLO, and results showed that the infection of SCRV can reduce the luciferase activity of pmirGLO-MDA5-exon1, and the use of cycloleucine (CL, an amino acid analogue that can inhibit m⁶A modification) hindered the degradation of MDA5-exon1 by SCRV, indicating that SCRV can regulate MDA5 expression through a methylation mechanism (*Figure 5G*). MDA5 is classified as an interferon-stimulated gene (ISG). Consistently, in spleen and kidney tissues infected with SCRV, the expression level of MDA5 demonstrates fluctuation. (*Figure 5H*). We further explored the mRNA and protein levels of MDA5 in MKC cells infected with SCRV and poly(I:C)-HMW and found that their expression all first significantly increased and then returned to the normal level (*Figure 5I and J*). Overall, our results suggested that the m⁶A modification of MDA5 is increased, and the expression level of MDA5 shows fluctuating changes upon SCRV infection.

## m⁶A-modification weakens MDA5 mRNA stability and antiviral ability

SCRV infection can increase the methylation level of MDA5, and this mechanism is expected to have a regulatory effect on MDA5. To investigate this, the regulatory effects of METTL3 and METTL14, the main components of the methyltransferase complex, on MDA5 were explored. Three siRNAs targeting METTL3 and METTL14 were designed, and si-METTL3-2 and si-METTL14-3 showed higher knockdown efficiency (*Figure 6A*). METTL3 and METTL14-overexpressing plasmids were also constructed, and the interaction between them confirmed their cooperative role in m⁶A modification (*Figure 6B*). Subsequently, the global m⁶A levels were measured in cells after knockdown or overexpression of METTL3 and METTL14. Consistent with expectations, the knockdown of METTL3 and METTL14 significantly decreased the global m⁶A level, while their overexpression significantly increased the overall methylation level (*Figure 6C*). To investigate their role in the process of virus infection, we used the virus plaque and qPCR methods to explore whether it could affect SCRV replication. As shown in (*Figure 6D and E*), overexpression of METTL3 or METTL14 significantly enhances viral plaque formation and promotes SCRV replication in SCRV-infected cells. And as m⁶A writers, METTL3 and METTL14 can directly regulate the methylation level of MDA5 and induce MDA5 degradation by reducing its mRNA stability (*Figure 6F–I*). We further inserted the MDA5-exon1 into pmirGLO and mVenus-C1 vector, and results showed that si-METTL3&14 can reduce the luciferase activity of pmirGLO-MDA5-exon1, and METTL3&14 could suppress the levels of GFP, further indicating the presence of methylation sites in MDA5-exon1 (*Figure 6—figure supplement 1A and D*). To investigate the potential m⁶A sites on MDA5-exon1, we constructed mutant MDA5-exon1 reporter plasmids (*Figure 5D* and *Figure 6—figure supplement 1B*). As expected, our results showed that the luciferase activity of wild-type pmirGLO-MDA5-exon1 reporter, but not of mutant pmirGLO-MDA5-exon1 reporter, was markedly increased upon METTL3&14 knockdown (*Figure 6—figure supplement 1C*). Furthermore, we found that the use of cycloleucine hindered the degradation of MDA5 by METTL3 and METTL14, indicating that these proteins regulate MDA5 expression through a methylation mechanism (*Figure 6J and K*). Additionally, MDA5 expression was degraded by METTL3&14 in both SCRV-infected and uninfected groups (*Figure 6L*). To investigate the impact of this mechanism on viral infection, we co-transfected METTL3&14 and MDA5 plasmids and found that METTL3&14 partially suppressed the antiviral immune benefit of MDA5 (*Figure 6M–O*). Taken together, our data demonstrate that METTL3&14-mediated m⁶A methylation of MDA5 inhibits its mRNA stability and expression levels, thus suppressing the antiviral capabilities of MDA5.

## Detailed m⁶A regulatory mechanism of MDA5

The m⁶A mechanism is jointly regulated by writers, erasers, and readers. The m⁶A modification of mRNA recruits m⁶A-binding proteins (readers) which have the function of recognizing m⁶A sites and accelerating mRNA decay. In this study, we hypothesized that YTHDF readers (YTHDF1, YTHDF2, and YTHDF3) may recognize m⁶A-modified MDA5 and destabilize it. We examined the binding ability of YTHDF proteins to MDA5 mRNA and found that YTHDF2 and YTHDF3 exhibited stronger binding compared to YTHDF1 (*Figure 7A*). We then designed two siRNAs targeting YTHDF1, YTHDF2, and YTHDF3 respectively, and selected si-YTHDF1-1, si-YTHDF2-si-2, si-YTHDF3-2 with better knockdown effects for subsequent experiments (*Figure 7B*). Results showed that the knockdown or overexpression of YTHDF2 and YTHDF3 led to the increase or decrease of the expression levels of MDA5, whereas YTHDF1 did not appear to influence MDA5 expression (*Figure 7C and D*, *Figure 7—figure*

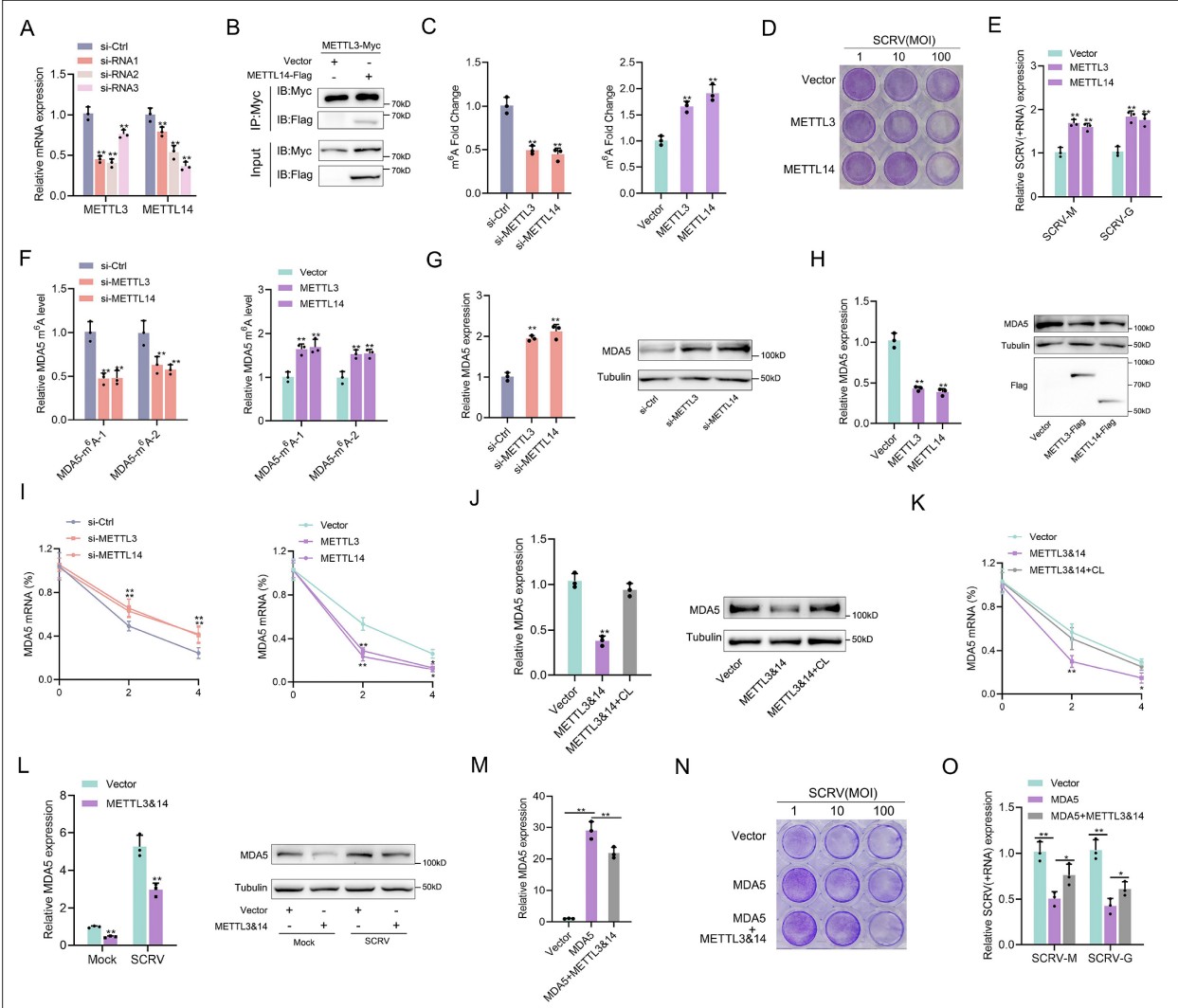

**Figure 6.** m[6]A-modification weakens MDA5 mRNA stability and antiviral ability. (**A**) Silence efficiency of si-METTL3 and si-METTL14 measured. Three siRNAs of METTL3 and METTL14 were transfected into MPC cells for 48 hr, respectively. (**B**) METTL3 immunoprecipitates with METTL14. EPC cells (1×10[7]) were co-transfected with METTL3-Myc and METTL14-Flag expression plasmids for 24 hr, followed by immunoprecipitation (IP) with anti-Myc. (**C**) METTL3 and METTL14 significantly increased the m[6]A content. MPCs were transfected with si-Ctrl or si-METTL3 or si-METTL14 and MKCs were transfected with vector or METTL3 or METTL14 plasmids for 48 hr, then the m[6]A level was measured by colorimetry. (**D**) METTL3 and METTL14 overexpressed MKC cells seeded in 48-well plates overnight were treated with SCRV at the dose indicated for 48 hr. Then, cell monolayers were fixed with 4% paraformaldehyde and stained with 1% crystal violet. (**E**) MKC cells were transfected with pcDNA3.1 vector and METTL3 or METTL14 expression plasmid for 24 hr, then infected with SCRV (MOI = 5) for 24 hr. The qPCR analysis was conducted for SCRV-M and SCRV-G RNA levels. (**F**) The m[6]A level alteration of MDA5 upon METTL3 or METTL14 knockdown or overexpression was examined by MeRIP-qPCR. MPC cells were transfected with si-Ctrl or si-METT3 and MKC cells were transfected with vector or METTL3 or METTL14 plasmids for 48 hr. (**G and H**) MPC cells were transfected with si-Ctrl, si-METTL3 or METTL14 and MKC cells were transfected with vector, METTL3, or METTL14 plasmids for 48 hr, the expression of MDA5 was detected by qRT-PCR and western blotting. (**I**) MPC cells were transfected with si-Ctrl, si-METTL3, or si-METTL14, and MKC cells were transfected with vector, METTL3, or METTL14 plasmids, then 5 μg/ml actinomycin D was added to the cells for 0 hr, 2 hr, and 4 hr. The half-life of MDA5 was analyzed by qRT-PCR. (**J**) MKC cells were transfected with vector or METTL3&14 plasmids for 24 hr and then treated with Cycloleucine (CL) for 24 hr in a final concentration of 20 mM. The expression of MDA5 was detected by qRT-PCR and western blotting. (**K**) MKC cells were transfected with vector or METTL3&14 plasmids for 24 hr and then treated with CL for 24 hr at 20 mM, then 5 μg/ml actinomycin D was added to the cells for 0 hr, 2 hr, and 4 hr. The half-life of MDA5 was analyzed by qRT-PCR. (**L**) MKC cells were transfected with vector or METTL3&14 plasmids for 24 hr and stimulated with SCRV (MOI = 5) for 24 hr, then the expression of MDA5 was detected by qRT-PCR and western blotting. (**M**) MKC cells seeded in 48-well plates overnight were transfected with MDA5 or MDA5 + METTL3&14 plasmids for 48 hr, then the expression of MDA5 was detected by qRT-PCR. (**N**) MKC cells seeded in 48-well plates overnight were transfected with MDA5 or MDA5 + METTL3&14 plasmids were treated with SCRV at the dose indicated for 48 hr. Then, cell monolayers were fixed with 4% paraformaldehyde and stained with 1% crystal violet. (**O**) MKC cells were transfected with MDA5 or MDA5 +

*Figure 6 continued on next page*

*Figure 6 continued*

METTL3&14 plasmids for 24 hr, then infected with SCRV (MOI = 5) for 24 hr. The qPCR analysis was conducted for SCRV-M and SCRV-G RNA levels. All data presented as the means ± SE from three independent triplicated experiments. **, p<0.01; *, p<0.05, as determined by Student's t test.

The online version of this article includes the following source data and figure supplement(s) for figure 6:

**Source data 1.** Excel files of data for *Figure 6*.

**Source data 2.** Raw unedited gels for *Figure 6*.

**Source data 3.** Uncropped and labeled gels for *Figure 6*.

**Figure supplement 1.** The regulation of m⁶A writers (METTL3&14) to the MDA5-exon1.

**Figure supplement 1—source data 1.** Excel files of data for *Figure 6—figure supplement 1*.

**Figure supplement 1—source data 2.** Raw unedited gels for *Figure 6—figure supplement 1*.

**Figure supplement 1—source data 3.** Uncropped and labeled gels for *Figure 6—figure supplement 1*.

*supplement 1A and C*). In addition, overexpression of YTHDF2&3 can increase the luciferase activity of wild-type pmirGLO-MDA5-exon1 reporter, but not of mutant pmirGLO-MDA5-exon1 reporter (*Figure 7—figure supplement 1B*). We then found that si-METTL3&14 can counteract the negative impact of YTHDF2&3 on MDA5 mRNA, protein, and stability (*Figure 7E and F*). Since the m⁶A mechanism is reversible, we then studied the regulation of MDA5 by two major erasers. Results showed that FTO, not ALKBH5, can upregulate MDA5 expression (*Figure 7G*). Moreover, the overexpression of FTO can reverse the inhibitory effect of YTHDF2&3 on MDA5 (*Figure 7H and I*). Taken together, these data suggested that through METTL3&14-m⁶A-YTHDF2&3 regulatory network, m⁶A can mediate the degradation of MDA5, and this process can be countered by FTO (*Figure 7J*).

## Discussion

In this study, we have identified multiple independent loss events of RIG-I throughout vertebrate evolution. Specifically, we have observed at least three independent loss events in mammals, birds, and fish, respectively. Regarding the loss of RIG-I in avian species, previous studies have provided more comprehensive and accurate information than ours, indicating that birds have experienced more than 16 instances of RIG-I loss (*Krchlíková et al., 2022*). However, it should be noted that we have not found the loss of RIG-I in reptiles and amphibians (not displayed). But this does not imply that there has been no loss event among them, but rather, there is insufficient information regarding the genomes of reptiles and amphibians to draw the conclusions. As the abundance of genomic data continues to grow, there will be an increasing amount of evidence available to verify the presence or absence of RIG-I in these taxonomic groups. Additionally, the presence of RIG-I in current species does not guarantee that RIG-I loss has not occurred previously. For instance, our research has indicated that zebrafish may have experienced RIG-I loss. However, prior to the loss event, the zebrafish RIG-I has undergone repetitive events. These findings collectively highlight that the loss of RIG-I is not a random occurrence, but rather bears significant biological and evolutionary implications that deserve further investigation.

The evolutionary loss of RIG-I raises the question of a possible compensation of its function by other gene products. RIG-I deficiency instances in mammals were scarcer compared to those in fish and birds. Previous research has revealed that in *T. belangeri*, concurrent with the loss of RIG-I, both MDA5 and LGP2 have undergone notable positive selection, and MDA5 or MDA5/LGP2 could sense Sendai virus (an RNA virus posed as a RIG-I agonist) for inducing type I IFN. This proposed an immune substitution model for RIG-I deficiency in mammals, though direct evidence of MDA5/LGP2 binding to 5'ppp-RNA was lacking. In this study, we characterized the consequences resulting from the loss of RIG-I in the *M. miiuy* and *G. gallus* (two representative species of vertebrates lacking RIG-I) and provided direct functional evidence for the alternative immune recognition function of MDA5. SCRV and VSV belongs to the class of rhabdovirus, which is a negative-strand virus with a non-segmented genome and initiate both replication and transcription de novo leading to 5'-triphosphate RNA in the cytosol, was expected to trigger an IFN response without the need for replication and presumed dsRNA formation (*Hornung et al., 2006*). Then, we synthesized 5'ppp-RNA probes from the SCRV and VSV genome sequences and purified recombinant MDA5 of *M. miiuy* and *G. gallus*. Results demonstrated that MDA5 detect the 5'ppp-RNA in *M. miiuy* and *G. gallus* and is disrupted when RNA

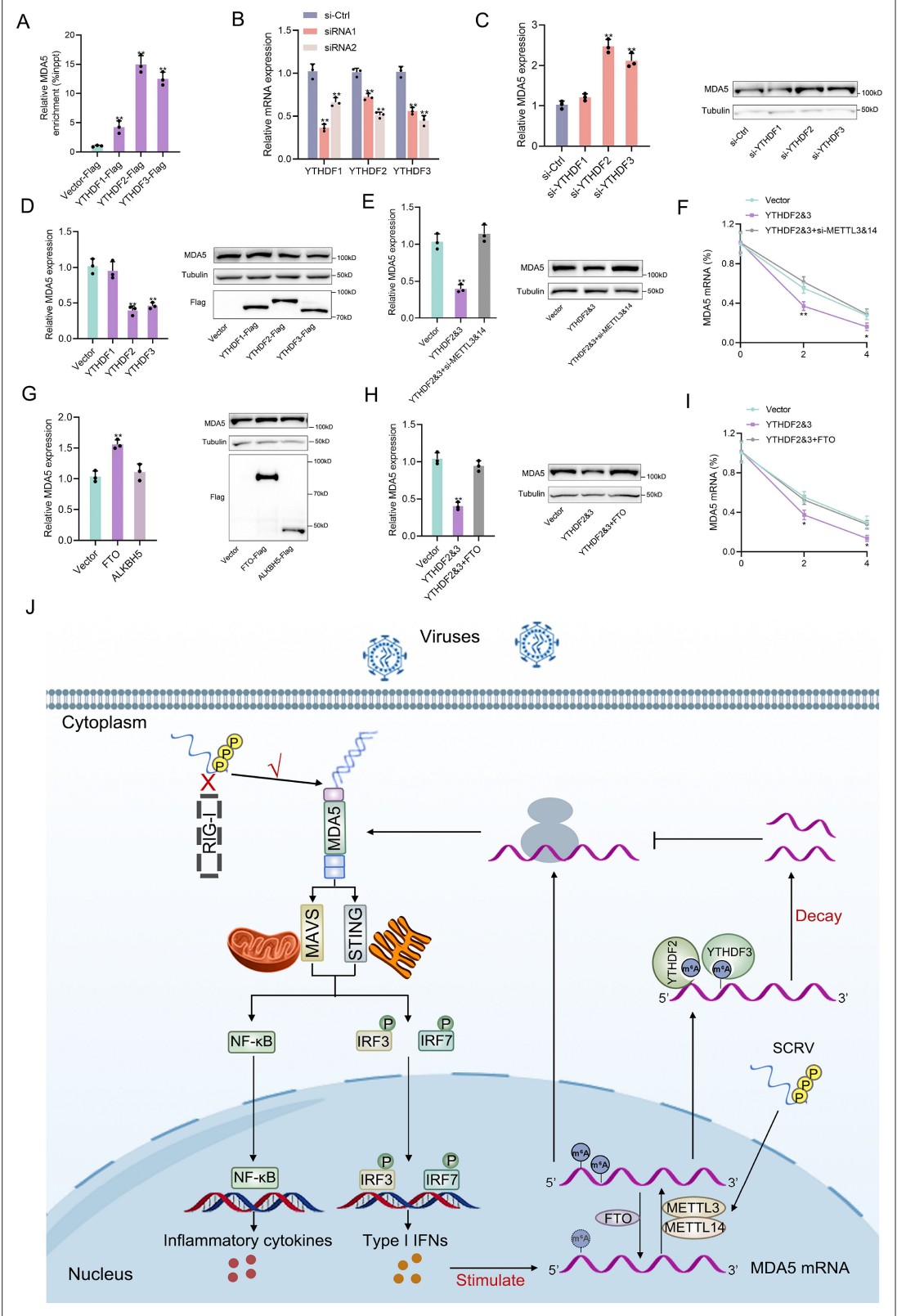

**Figure 7.** Detailed m⁶A regulatory mechanism of MDA5. (**A**) The binding relationship between YTHDF1, YTHDF2, or YTHDF3 and MDA5 mRNA was validated using a RIP assay. MKC cells were transfected with YTHDF1-Flag, YTHDF2-Flag, YTHDF3-Flag, or pcDNA3.1-Flag for 48 hr. (**B**) SiRNA silencing effect test of YTHDF1, YTHDF2, and YTHDF3. MPC cells were transfected with si-YTHDF1, si-YTHDF2, or si-YTHDF3 for 48 h. (**C and D**) MPC cells were transfected with si-YTHDF1, si-YTHDF2, si-YTHDF3 or si-Ctrl (**C**), and MKC cells were transfected with YTHDF1, YTHDF2, YTHDF3 or vector for 48 hr

*Figure 7 continued on next page*

*Figure 7 continued*

(**D**), then the expression of MDA5 was detected by qRT-PCR and western blotting. (**E**) Relative mRNA and protein levels of MDA5 in MKC cells after co-transfected with YTHDF2&3 plasmids and si-METTL3&14 by qPCR and western blot assays. (**F**) MKC cells were transfected with YTHDF2&3 plasmids and si-METTL3&14, then 5 μg/ml actinomycin D was added to the cells for 0 hr, 2 hr, and 4 hr. The half-life of MDA5 was analyzed by qRT-PCR. (**G**) MKC cells were transfected with vector, FTO, ALKBH5 plasmids for 48 hr, then the expression of MDA5 was detected by qRT-PCR and Western blotting. (**H**) Relative mRNA and protein levels of MDA5 in MKC cells after co-transfected with YTHDF2&3 plasmids and FTO plasmids by qPCR and western blotting. (**I**) MKC cells were transfected with YTHDF2&3 plasmids and FTO plasmids, then 5 μg/ml actinomycin D was added to the cells for 0 hr, 2 hr, and 4 hr. The half-life of MDA5 was analyzed by qRT-PCR. (**J**) Schematic diagram of arms race between MDA5 and 5'ppp-RNA virus in *M. miiuy*. All data presented as the means ± SE from three independent triplicated experiments. **, $p<0.01$; *, $p<0.05$, as determined by Student's t test.

The online version of this article includes the following source data and figure supplement(s) for figure 7:

**Source data 1.** Excel files of data for *Figure 7*.

**Source data 2.** Raw unedited gels for *Figure 7*.

**Source data 3.** Uncropped and labeled gels for *Figure 7*.

**Figure supplement 1.** The regulation of YTHDF readers (YTHDF1&2&3) to the MDA5-exon1.

**Figure supplement 1—source data 1.** Excel files of data for *Figure 7—figure supplement 1*.

**Figure supplement 1—source data 2.** Raw unedited gels for *Figure 7—figure supplement 1*.

**Figure supplement 1—source data 3.** Uncropped and labeled gels for *Figure 7—figure supplement 1*.

is capped at the 5'-triphosphate end or dephosphorylated. Furthermore, the recognition of RNA by MDA5 depends on its RD domain, which not only indicates that the RD domain of MDA5 has evolved functions similar to RD domain of RIG-I, but also excludes the possibility of MDA5 binding to 5'ppp-RNA through intermediate ligands. Furthermore, in vertebrates that possess RIG-I, STING exclusively binds to RIG-I rather than MDA5. However, similar to the interaction observed between MDA5 and STING in tree shrews and chickens that lacking RIG-I (*Xu et al., 2016*; *Xu et al., 2019*), the MDA5 of M. miiuy can also interact with STING protein, indicating that in vertebrates lacking RIG-I, MDA5 may utilize STING to facilitate signal transduction in the antiviral response. Next, by linking the immune replacement ability evolved by MDA5 with the common RIG-I loss event in vertebrates, we propose a straightforward explanation for this evolutionary event: In the lengthy process of vertebrate evolution, MDA5 gradually acquired new functionalities to effectively detect viruses such as 5'ppp-RNAs, which were initially recognized by RIG-I. As a result, there was functional redundancy of RIG-I, which eventually led to its gradual disappearance from the vertebrate genome. This hypothesis is supported by two viewpoints: first, MDA5 and RIG-I can both recognize overlapping viruses *Schoggins et al., 2015*; secondly, in fish without RIG-I deficiency, MDA5 has stronger antiviral ability compared to RIG-I (*Wan et al., 2017*). Finally, although MDA5 has a compensatory effect on RIG-I, the absence of RIG-I may still have a significant impact on the immune system of vertebrates. Ducks are often resistant to influenza viruses capable of killing chicken, and the loss of RIG-I in chickens provides a plausible explanation for their increased susceptibility to influenza viruses compared with ducks (*Barber et al., 2010*). Furthermore, SCRV virus, the virus model used in this study, mainly affects the Siniperca chuatsi, which is a fish species that also lacks RIG-I like *M. miiuy* (*Liu et al., 2023*).

Viruses are constantly co-evolving with hosts. Our available data suggested that the vertebrate has evolved MDA5 with alternative functions to recognize and resist 5'ppp-RNA virus in the absence of RIG-I. As a response, virus has also evolved to use the m⁶A mechanism to evade the host immune response. Previous study indicated that m⁶A modification plays a crucial role in maintaining immune homeostasis in the body (*Gan et al., 2023*). During the infection process, the virus may disrupt the m⁶A level and disrupt the immune balance, thereby facilitating its own replication and invasion. Viruses may utilize the m⁶A mechanism in two ways to evade immunity. On the one hand, many studies suggested a role for m⁶A in shielding RNA species from detection by PRRs. There are numerous viruses recognized in RIG-Idependent manner, but m⁶A modification increases the possibility of immune evasion. For example, the loss of m⁶A site or removing m⁶A from human metapneumovirus may weaken its immune evasion ability, prevent it from evading detection by RIG-I, and result in higher levels of IFN release (*Lu et al., 2020*). Moreover, it has been reported that the RNA containing m⁶A modifications binds to RIG-I poorly, and it failed to trigger the RIG-I conformational changes associated with downstream immune signals (*Durbin et al., 2016*). On the other hand, m⁶A-mediated viral strategies for inhibiting IFN induction pathways appear to exist. For instance, following infection by human

cytomegalovirus (HCMV), the IFN-β transcript was modified by m⁶A, and this methylation decreased its half-life, suggesting that HCMV can exploit this control of IFN-β expression to facilitate its replication by increasing m⁶A modification of the IFN-β transcript and thus decreasing its production (**Rubio et al., 2018**). In another study, the m6A reader YTHDF3 could restrain the type I IFN response and ISG expression by involving enhancement of FOXO3 translation in a m6A-independent mechanism, which acts as a negative regulator of immunity (**Zhang et al., 2019**). Moreover, previous study reports that IFN-α and IFN-β mRNA encoding cytokines that promote IFN response are modified by m6A, and the loss of METTL3 and YTHDF2 could increase the IFN expression and ISG activity (**Winkler et al., 2019**). In summary, viruses can utilize m⁶A to evade the immune response by shielding the recognition of RIG-I or reducing the host's immune reaction, thus counteracting the host's defense mechanisms. In our previous study, we discovered that SCRV infection can not only increase overall cellular methylation levels but also upregulate the expression of METTL3, thereby suggesting its potential to regulate host gene expression through the m⁶A mechanism (**Geng et al., 2023**). While changes in m⁶A modification and the expression of m⁶A-modified transcripts are biologically relevant, identifying bona fide m⁶A alterations during viral infection will allow us to understand how m⁶A modification of cellular mRNA is regulated. As our m⁶A change analysis pipeline controls for changes in gene expression, these data should represent true changes in m⁶A modification rather than changes in the expression of m⁶A-modified transcripts. In this study, we found that SCRV virus infection alters m⁶A modification of specific cellular transcripts, including MDA5. MDA5 plays a crucial role as a receptor in recognizing the SCRV virus and controlling IFN activation. The aberrant m⁶A levels of MDA5 subsequently affect the expression of antiviral genes, thereby compromising its ability to mount an effective antiviral immune response. It is plausible that SCRV also impacts the production of antiviral genes, such as IFN-1 or other cytokines, by manipulating m⁶A modification on the transcripts of cytokines or molecules involved in their production. This interesting aspect warrants further investigation in future research.

In this study, we demonstrated a co-evolutionary arms race between 5'ppp-RNA virus and virus receptors. Specifically, we conducted an evolutionary analysis and provided functional evidence to confirm that MDA5 of M. miiuy and G. gallus has acquired an additional function of sensing 5'ppp-RNA, thus compensating for the loss of RIG-I. Additionally, we found that SCRV infection can regulate the m⁶A level of MDA5 in the M. miiuy, leading to its degradation and subsequently affecting the immune response for immune evasion. Fish and birds, as the most abundant aquatic and terrestrial vertebrates respectively, are also the two lineages with the most frequent occurrence of RIG-I deficiency, the investigation of host-pathogen interactions of them offers valuable insights into the ecological and evolutionary factors that contribute to the diversity of the immune system.

## Materials and methods

### Key resources table

| Reagent type (species) or resource | Designation | Source or reference | Identifiers | Additional information |
|---|---|---|---|---|
| Gene (*M. miiuy*) | MDA5 | GenBank | PP179381.1 | |
| Gene (*M. miiuy*) | LGP2 | GenBank | KX351161.1 | |
| Gene (*G. gallus*) | MDA5 | GenBank | NM_001193638.2 | |
| Strain, strain background (*siniperca cheats* rhabdovirus) | SCRV virus | This paper | Materials and methods: Fish and challenge | Isolated from mandarin fish |
| Strain, strain background (Vesicular Stomatitis Virus) | VSV virus | BrainCase | VSV31 | |
| Cell line (*M. miiuy*) | MKC | This paper | Materials and methods: Cell culture | *M. miiuy* kindey cell line |
| Cell line (*M. miiuy*) | MPC | This paper | Materials and methods: Cell culture | *M. miiuy* spleen cell line |

*Continued on next page*

*Continued*

| Reagent type (species) or resource | Designation | Source or reference | Identifiers | Additional information |
|---|---|---|---|---|
| Cell line (*G. gallus*) | DF-1 | ATCC | Cat# MZ-2647 RRID:CVCL_0570 | |
| Cell line (*Cyprinus carpio*) | EPC | ATCC | Cat# CRL-2872 RRID:CVCL_4361 | |
| Cell line (*Homo sapiens*) | HEK293T | Beyotime | Cat# C6008 RRID:CVCL_0063 | |
| Biological sample (*M. miiuy*) | Kidney tissue | This paper | Materials and methods: Fish and challenge | Isolated from *M. miiuy* |
| Biological sample (*M. miiuy*) | Spleen tissue | This paper | Materials and methods: Fish and challenge | Isolated from *M. miiuy* |
| Antibody | Anti-MDA5 (Rabbit polyclonal) | Beyotime | Cat# AF7164 | WB (1:500) |
| Antibody | Anti-IRF3 (Rabbit polyclonal) | Boster | Cat# BA4351-2 | WB (1:500) |
| Antibody | Anti-Flag (Mouse polyclonal) | Beyotime | Cat# AF519 | IF (1:300), WB (1:1000) |
| Antibody | Anti-Myc (Mouse polyclonal) | Beyotime | Cat# AF2864 | WB (1:1000) |
| Antibody | Anti-HA (Mouse polyclonal) | Beyotime | Cat# AF2858 | WB (1:1000) |
| Antibody | Anti-Tubulin (Mouse polyclonal) | Beyotime | Cat# AT819 | WB (1:1000) |
| Antibody | Anti-His (Mouse polyclonal) | Beyotime | Cat# AT819 | WB (1:1000) |
| Antibody | HRP-conjugated anti-rabbit IgG | Abbkine | Cat# A25022 | WB (1:5000) |
| Antibody | HRP-conjugated anti-mouse IgG | Abbkine | Cat# A25012 | WB (1:5000) |
| Antibody | Anti-GFP (Mouse monoclonal) | Beyotime | Cat# AG281 | WB (1:1000) |
| Recombinant DNA reagent | PcDNA3.1 | Invitrogen | Cat# V79020 | |
| Recombinant DNA reagent | pmirGLO vector | Promega | Cat# E1330 | |
| Sequence-based reagent | PCR Primers | Synthesized in Genewiz | Listed in **Supplementary file 1** | |
| Sequence-based reagent | siRNAs | Synthesized in GenePharma | Listed in section 4.7 | |
| Peptide, recombinant protein | Rnase A | Beyotime | Cat# ST579 | |
| Peptide, recombinant protein | Dnase I | Beyotime | Cat# D7076 | |
| Peptide, recombinant protein | Calf Intestinal Alkaline Phosphatase (CIAP) | Invitrogen | Cat# 18009–019 | |
| Peptide, recombinant protein | Ribo RNAmax-T7 kit | RiboBio | Cat# C11001-2 | |
| Commercial assay or kit | PureBindingRNA-Protein pull-down Kit | Geneseed | Cat# P0202 | |
| Commercial assay or kit | RNA EMSA kit | Beyotime | Cat# GS606 | |
| Commercial assay or kit | Magna RIP RNA-Binding Protein Immunoprecipitation Kit | Millipore | Cat# 17–700 | |
| Commercial assay or kit | m6A RNA Enrichment Kit | Epigentek | Cat# P-9018–24 | |
| Commercial assay or kit | BCA Protein Assay kit | Beyotime | Cat# P0012S | |
| Commercial assay or kit | Endotoxin-Free Plasmid DNA Miniprep Kit | Tiangen | Cat# DP118 | |

*Continued on next page*

*Continued*

| Reagent type (species) or resource | Designation | Source or reference | Identifiers | Additional information |
|---|---|---|---|---|
| Commercial assay or kit | Dual-Luciferase Reporter Assay System | Promega | Cat# E1980 | |
| Commercial assay or kit | FastQuant RT Kit | Tiangen | Cat# KR106-03 | |
| Commercial assay or kit | SYBR Premix Ex Taq | Takara | Cat# DRR041S | |
| Commercial assay or kit | Lipofectamine RNAiMAX | Invitrogen | Cat# 13778150 | |
| Commercial assay or kit | Lipofectamine 3000 | Invitrogen | Cat# L3000015 | |
| Commercial assay or kit | Ribo RNAmax-T7 Biotin Labeling Transcription Kit | RiboBio | Cat# C11002-2 | |
| Chemical compound, drug | Poly(I:C)-HMW | invivogen | Cat# 31852-29-6 | 10 µg/ml |
| Chemical compound, drug | DAPI | Beyotime | Cat# C1002 | |
| Chemical compound, drug | m7G cap analogue | Yeason | Cat# 10678ES10 | |
| Chemical compound, drug | Cycloleucine | Macklin | Cat# C830431 | 20 mM |
| Chemical compound, drug | Actinomycin D | Solarbio | Cat# A8030-5 | 5 µg/ml |
| Software, algorithm | GraphPad Prism 8 | GraphPad Software | RRID:SCR_002798 | |
| Software, algorithm | TBtools | TBtools | RRID:SCR_023018 | |
| Other | Anti-His beads | Solarbio | Cat# M2300 | Used for protein purification |

## Fish and challenge

*M. miiuy* (~50 g) was obtained from Zhoushan Fisheries Research Institute, Zhejiang Province, China. Fish was acclimated in aerated seawater tanks at 25°C for 6 weeks before experiments. SCRV virus was isolated from mandarin fish as described previously (*Liu et al., 2023*). The experimental procedure for SCRV infection was performed as described (*Chu et al., 2020*). Briefly, In the SCRV injection group, fish was challenged with 200 µl SCRV at a multiplicity of infection (MOI) of 5 through intraperitoneal. As a comparison, 200 µl of physiological saline was used to challenge the individuals. Afterwards, fishes were respectively sacrificed at different time point and the kidney and spleen tissues were collected for RNA extraction.

## Cell culture

*M. miiuy* kidney cells (MKCs) and spleen cells (MPCs) are cell lines derived from the kidney and spleen tissues of *M. miiuy* (*Geng et al., 2022*), with passages ranging from 20 to 40 times. MKC and MPC cells were cultured in L-15 medium (HyClone) supplemented with 15% fetal bovine serum (FBS; Gibco), 100 U/ml penicillin, and 100 µg/ml streptomycin at 26°C. Fish EPC cells (Epithelioma papulosum cyprini cells) were maintained in medium 199 (Invitrogen) supplemented with 10% FBS, 100 U/ml penicillin, and 100 mg/ml streptomycin at 26°C in 5% $CO_2$. HEK293T cells were cultured in DMEM (HyClone) supplemented with 10% FBS, 100 U/ml penicillin, and 100 mg/ml streptomycin at 37°C in 5% $CO_2$. Chicken embryo fibroblast cells (DF-1) were cultured in DMEM (HyClone) supplemented with 10% FBS, 100 U/ml penicillin, and 100 mg/ml streptomycin at 39°C in 5% $CO_2$.

## Oligonucleotides

The sequences of the 67 nt 5'ppp-SCRV or 5'ppp-VSV were derived from the 5' and 3'UTRs of the VSV or SCRV genome as previously described (*Goulet et al., 2013*), and the sequence of 112 bp dsRNA was taken from the first 100 bp of the MDA5 gene of miiuy croaker flanked by 5'-gggaga and tctccc-3'. These RNAs were synthesized by in vitro transcription using the Ribo RNAmax-T7 kit (RiboBio). Biotin-labeled RNAs were transcribed in vitro using a Ribo RNAmax-T7 Biotin Labeling Transcription Kit (RiboBio). For generating duplexes, RNA oligonucleotides were mixed in hybridization buffer (20 mM Tris-HCl [pH 8.0], 1.5 mM $MgCl_2$, and 1.5 mM DTT), boiled for 1 min, and incubated at 37°C for 1 hr. For removal of 5'-triphosphates, 20 mg of RNA was treated with 20 U Calf Intestinal Alkaline Phosphatase (CIAP, Invitrogen) for 2 hr at 37°C in the presence of RNase inhibitor (Beyotime)

and extracted with phenolchloroform. RNA molecules with a terminal 5' m7G cap were synthesized by the incorporation of m7G cap analogue (Yeason) in the transcription reactions. RNA was analysed on a denaturing 17% polyacrylamide, 7 M urea gel following digestion with 50 ng/μl of RNase A (Beyotime) or 100 mU/μl of DNase I (Beyotime) for 30 min.

## Plasmids construction

To construct Flag or Myc-tagged MDA5 (GeneBank: PP179381.1), MAVS, STING, METTL3, METTL14, FTO, ALKBH5, YTHDF1, YTHDF2, and YTHDF3 expression plasmids, their CDS sequences of the *M. miiuy* were amplified by PCR and then ligated into a pcDNA3.1 vector (Invitrogen), respectively. Then, MDA5-△RD were generated by PCR on the basis of MDA5 plasmid. To construct MDA5-exon1 reporter vector, the exon1 region of *M. miiuy* MDA5 gene was amplified using PCR with the gene-specific primers and ligated into a pmir-GLO luciferase reporter vector (Promega). Meanwhile, *M. miiuy* MDA5-exon1 were inserted into the mVenus-C1 (Invitrogen), which included the sequence of enhanced GFP. For mutant reporter vectors (MDA5-exon1-mut), adenosine (A) in the predicted m$^6$A motif was replaced by an uracil (U) by using the Mut Express II Fast Mutagenesis Kit V2 (Vazyme) with specific primers. To construct Flag-tagged MDA5 expression plasmid of *G. gallus*, the CDS sequence of MDA5 was amplified by PCR and then ligated into a pcDNA3.1 vector (Invitrogen). The correct construction of the recombinant plasmids was verified through Sanger sequencing and extracted using an Endotoxin-Free Plasmid DNA Miniprep Kit (Tiangen, China) before using plasmids. The primers were listed in *Supplementary file 1*.

## Protein purification and analysis

For protein purification, MDA5 plasmids with 6×His tag was constructed based on pcDNA3.1. (His6)-Flag-tagged MDA5 of *M. miiuy* and *G. gallus* were transiently overexpressed in 293T cells and lysed in a binding buffer (500 mM NaCl, 20 mM Tris/HCl [pH 7.4], 20 mM Imidazole, and 1% Triton X-100) including protease inhibitor cocktail (Roche). The lysate was incubated over night at 4°C with anti-His beads (Solarbio). Anti-His beads were washed subsequently with washing buffer (500 mM NaCl, 20 mM Tris/HCl [pH 7.4], 20 mM Imidazole, and 1% Triton X-100). MDA5 proteins were eluted by an addtion of elution buffer (500 mM NaCl, 20 mM Tris/HCl [pH 7.4], 500 mM Imidazole, and 1% Triton X-100) to the beads. Purity of recombinant MDA5 was determined by SDS-PAGE separation.

## RNA interference

The MDA5-specific small interfering RNA (si-MDA5-1 and si-MDA5-2) sequences were 5'- CGGACUAC AUGCAGCGUAATT-3' and 5'-GACCAAUGAGAUUUCUAUGTT-3', respectively. The LGP2-specific small interfering RNA (si-LGP2) sequence was 5'-GGUUCACCUUGUAGAGCAATT-3'. The STING-specific small interfering RNA (si-STING) sequence was 5'-ACGACAGCAUCCAUUUCUATT-3'. The METTL3-specific small interfering RNA (si-METTL3-1, si-METTL3-2, and si-METTL3-3) sequences were 5'-CAUCACAAACGAACUCAACTT-3', 5'- AACGUGGGCAAACUCUUUUTT-3' and 5'-GAGAUUCU GGAACUACUUAUT-3', respectively. The METTL14-specific small interfering RNA (si-METTL14-1, si-METTL14-2, and si-METTL14-3) sequences were 5'-GCUGGAAAUCGAGGAGAUAUT-3', 5'- CCGT ACGAAGAGGTGTACATT-3' and 5'-GAGCCUCCCUUGGAAGAAUTT-3', respectively. The YTHDF1-specific small interfering RNA (si-YTHDF1-1 and si-YTHDF1-2) sequences were 5'-AAAGACUUUGAC UGGAACUTT-3' and 5'-CAGUCGAUCAGAGACCUAATT-3', respectively. The YTHDF2-specific small interfering RNA (si-YTHDF2-1 and si-YTHDF2-2) sequences were 5'-GCAGGGUGUUUAUCAUCAAT T-3', 5'- CUAUGCUCCCAGCUCAAUUTT-3', respectively. The YTHDF3-specific small interfering RNA (si-YTHDF3-1 and si-YTHDF3-2) sequences were 5'-GGUGGACUACAAUGCCUAUTT-3' and 5'-UCUA CAGUAACAGCUAUGGTT-3', respectively. The scrambled control RNA sequences were 5' - UUCU CCGAACGUGUCACGUTT - 3'. They were purchased from GenePharma (Shanghai, China).

## Cells transfection

Transient transfection of cells with siRNA was performed in 24-well plates using Lipofectamine RNAiMAX (Invitrogen), and cells were transfected with DNA plasmids was performed using Lipofectamine 3000 (Invitrogen) according to the manufacturer's instructions. For functional analyses, the overexpression plasmid (500 ng per well) or control vector (500 ng per well) and siRNA (100 nM) were transfected into cells in culture medium and then harvested for further detection.

## RNA extraction and quantitative real-time PCR

Total RNA was extracted using the TRIzol Reagent (Invitrogen) and the cDNA was synthesized using the FastQuant RT Kit (Tiangen) which contains DNase treatment of RNA to eliminate genomic contamination, following the manufacturer's instructions. The expression profiles of each gene were conducted by using the SYBR Premix Ex Taq (Takara), as previously described. The quantitative real-time PCR was conducted in an Applied Biosystems QuantStudio 3 (Thermo Fisher Scientific). β-actin was used as internal controls. Primer sequences are listed in *Supplementary file 1*.

## Western blotting

Cellular lysates were generated by using 1×SDS PAGE loading buffer. Proteins were extracted from cells and measured with the BCA Protein Assay kit (Vazyme), then subjected to SDS-PAGE (10%) gel and transferred to PVDF (Millipore) membranes by semidry blotting (Bio-Rad Trans Blot Turbo System). The membranes were blocked with 5% BSA. Protein was blotted with different antibodies. The antibody against MDA5 was diluted at 1: 500 (Beyotime, Cat# AF7164); The antibody against IRF3 was diluted at 1: 500 (Boster, Cat# BA4351-2); anti-Flag, anti-Myc, anti-HA and anti-Tubulin monoclonal antibody were diluted at 1: 1,000 (Beyotime); and HRP-conjugated anti-rabbit IgG or anti-mouse IgG (Abbkine) at 1: 5,000. The results were the representative of three independent experiments. The immunoreactive proteins were detected by using WesternBright ECL (Advansta). The digital imaging was performed with a cold CCD camera.

## Dual-luciferase reporter assays

For luciferase assays, MKC cells were co-transfected MDA5 expression plasmid, together with NF-κB, IRF3, IRF7, or IFN-1 luciferase reporter genes, and the pRL-TK *Renilla* luciferase reporter plasmid. For the detection of m⁶A site, the wild or mutant of MDA5-exon1 luciferase reporter was co-transfected si-METTL3, si-METTL14, si-YTHDF1, si-YTHDF2, si-YTHDF3 or si-Ctrl into MKC cells. After 48 hr transfection, the cells were collected and lysed for the reporter luciferase activities measured by using a dual-luciferase reporter assay system (Promega). All the luciferase activity values were gained against the *Renilla* luciferase control.

## RNA binding protein immunoprecipitation (RIP)

To identify whether MDA5 or MDA5-△RD can directly bind to SCRV viral RNA, MKC cells (~2.0 × 10⁷) were transfected with pcDNA3.1-Flag, pcDNA3.1-MDA5-Flag or pcDNA3.1-MDA5-△RD-Flag and then infected with SCRV at an MOI of 5 for 24 hr. In addition, to identify whether YTHDF1, YTHDF2, and YTHDF3 can directly bind to MDA5, MKC cells were co-transfected with pcDNA3.1-Flag, pcDNA3.1-YTHDF1/2/3-Flag. For RNA immunoprecipitation (RIP) assays, MKC cells were harvested after 48 hr transfection, and RIP assays were carried out with Magna RIP RNA-Binding Protein Immunoprecipitation Kit (Millipore) and anti-Flag antibody (Abcam) following the manufacturer's protocol. Then, the expression of target gene SCRV-M, SCRV-G, β-actin, or MDA5 were detected by qRT-PCR analysis.

## RNA pull-down assay

Cytoplasmic extracts were prepared from 2×10⁷ MKC cells transfected with mmiMDA5-Flag (MDA5 of miiuy croaker) plasmids and 5×10⁷ DF1 cells transfected with gga-MDA5-Flag (MDA5 of chicken) plasmids. The extracts were incubated with biotinylated 5'ppp-RNA or dsRNA and subjected to pull-down with streptavidin agarose beads (Geneseed), followed by subsequent Coomassie bluestaining or SDS-PAGE analysis and immunoblotting with anti-Flag antibody (Beyotime).

## EMSA assay

Recombinant MDA5 proteins were mixed with biotin-labeled oligonucleotides in a reaction mixture (10 μl: 20 mM Tris-HCl pH 8.0, 1.5 mM MgCl₂, 1.5 mM DTT). After incubation at room temperature for 15 min, the reaction mixtures were conducted on 6% non-denaturing polyacrylamide gels and the EMSA was performed using Chemiluminescent EMSA Kit (Beyotime).

## MeRIP-seq assays and analysis

Total RNA was isolated using TRIzol reagent. m⁶A immunoprecipitation and library preparation were performed following the manufacturer's protocol. Poly (A) RNA is purified from 50 μg total RNA using

Dynabeads Oligo (dT) (Thermo Fisher) using two rounds of purification. Then the poly(A) RNA was fragmented into small pieces using Magnesium RNA Fragmentation Module (NEB) under 86°C for 7 min. Then the cleaved RNA fragments were incubated for 2 hr at 4°C with m6A-specific antibody (Synaptic Systems) in IP buffer (50 mM Tris-HCl, 750 mM NaCl and 0.5% Igepal CA-630). For sequencing data analysis, we used HISAT2 to map reads to the reference genome of *M. miiuy*. Mapped reads of IP and input libraries were provided for R package exomePeak, which identifies m6A peaks with bed or bigwig format that can be adapted for visualization on the IGV software. The differentially expressed mRNAs were selected with log2 (fold change)>1 or log2 (fold change) <-1 and p-value <0.05 by R package edgeR. The circus map of m6A peaks in untreated and SCRV-infected spleen tissues of *M. miiuy* was showed by TBtools (*Chen et al., 2020*).

## MeRIP-qPCR

Total RNA was extracted using TRIzol reagent (Invitrogen). Methylated m6A RNA immunoprecipitation (Me-RIP) was performed according to the protocol of EpiQuik CUT&RUN m6A RNA Enrichment Kit (Epigentek). qPCR analysis of the methylated RNA was performed to detect methylated MDA5 mRNA levels. Relative m6A level for each transcript was calculated as the percent of input in each condition normalized to that of the respective positive control spike-in. Fold change of enrichment was calculated with mock samples normalized to 1 (*McIntyre et al., 2020*).

## Total m6A quantification assay

Total RNA was extracted by the TRIzol Reagent (Invitrogen). Then, an EpiQuik m6A RNA Methylation Quantification Kit (EpiGentek) was used to detect the total m6A level according to the manufacturer's protocol. Briefly, positive control (PC), negative control (NC), and 200 ng isolated mRNA were added to each well with the capture antibody. Next, the RNA was incubated with m6A antibody in wells, and m6A content was quantified at a wavelength of 450 nm.

## RNA stability assays

Cells were treated with actinomycin D (5 μg/mL) and then collected at different time points. RNA was extracted by TRIzol reagent (Invitrogen), and the mRNA levels were measured using qRT-PCR.

## Database mining and sequence analysis

The species tree is constructed by submitting species names to the NCBI (https://www.ncbi.nlm.nih.gov/Taxonomy/CommonTree/wwwcmt.cgi). In order to determine whether there is RIG-I in vertebrates, we use *H. sapiens* and *D. rerio* RIG-I protein sequences to TBLASTN the whole genome sequence of the species on the ensemble website (http://www.ensemble.org/). In addition, in order to prevent the possibility of incomplete species genome, we also used RIG-I protein sequences of *H. sapiens* and *D. rerio* to compare the transcriptome and genome sequenced of the designated species on NCBI website (http://www.ncbi.nlm.nih.gov/Genbank/). For further gene synteny analysis, RIG-I of *H. sapiens*, *C. milii*, *D. rerio* were used as anchor sites. In order to identify the MDA5 and LGP2 from *M. miiuy*, we used the homologues of zebrafish reported previously as queries to seek for the transcriptome (*Che et al., 2014*) and a chromosome-scale genome database *Xu et al., 2024* of *M. miiuy* by using local TBLASTN and BLASTN programs, then the protein structures were predicted by SMART website (http://smart.embl-heidelberg.de/).

## Statistical analysis

Data are expressed as the mean ± SE from at least three independent triplicated experiments. A two-sided Student's t test was used to evaluate the data. The relative gene expression data was acquired using the $2^{-\Delta\Delta CT}$ method, and comparisons between groups were analyzed by one-way analysis of variance (ANOVA) followed by Duncan's multiple comparison tests. A value of p<0.05 was considered significant.

## Acknowledgements

The authors are very grateful to the members of the Xu lab for their help and support. The authors would especially like to thank Wenhui Li and Ruotong Tian for their help with the construction of initial plasmids. Funding National Natural Science Foundation of China (31822057) Tianjun Xu. The funders

had no role in study design, data collection and interpretation, or the decision to submit the work for publication.

## Additional information

### Funding

| Funder | Grant reference number | Author |
| --- | --- | --- |
| National Natural Science Foundation of China | 31822057 | Tianjun Xu |

The funders had no role in study design, data collection and interpretation, or the decision to submit the work for publication.

### Author contributions

Shang Geng, Conceptualization, Data curation, Validation, Investigation, Writing – original draft; Xing Lv, Validation, Investigation; Weiwei Zheng, Resources, Supervision, Investigation; Tianjun Xu, Conceptualization, Supervision, Project administration, Writing – review and editing

### Author ORCIDs

Shang Geng (ID) https://orcid.org/0000-0003-0914-1048
Tianjun Xu (ID) https://orcid.org/0000-0003-3606-8069

### Ethics

All animal experimental procedures were performed in accordance with the National Institutes of Health's Guide for the Care and Use of Laboratory Animals, and the experimental protocols were approved by the Research Ethics Committee of Shanghai Ocean University (No. SHOU-DW-2018-047).

Reviewer #1 (Public review): https://doi.org/10.7554/eLife.94898.4.sa1
Reviewer #2 (Public review): https://doi.org/10.7554/eLife.94898.4.sa2
Author response https://doi.org/10.7554/eLife.94898.4.sa3

## Additional files

### Supplementary files

• Supplementary file 1. PCR primer information in this study.

• MDAR checklist

### Data availability

All data generated or analysed during this study are included in the manuscript and supporting files; source data files have been provided for Figures 1, 2,3,4,5,6,7, Figure 6—figure supplement 1, and Figure 7—figure supplement 1.

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
