## [Editor Report · eLife assessment]

This **important** study shows that in teleost fish, the RIG-I-like protein MDA5 can compensate for the absence of RIG-I by detecting 5'-triphosphorylated RNA. A fish virus containing such RNA can nevertheless evade MDA5 detection through a mechanism involving m6A methylation-induced silencing. The conclusions, which are supported by **solid** data, advance our understanding of antiviral immunity and virus-host conflicts in vertebrates.

---

## [Referee Report · Reviewer #1 (Public review)]

This study offers valuable insights into host-virus interactions, emphasizing the adaptability of the immune system. Readers should recognize the significance of MDA5 in potentially replacing RIG-I and the adversarial strategy employed by 5'ppp-RNA SCRV in degrading MDA5 mediated by m6A modification in different species, further indicating that m6A is a conservational process in the antiviral immune response.

However, caution is warranted in extrapolating these findings universally, given the dynamic nature of host-virus dynamics. The study provides a snapshot into the complexity of these interactions, but further research is needed to validate and extend these insights, considering potential variations across viral species and environmental contexts.

---

## [Referee Report · Reviewer #2 (Public review)]

Panel 2N and 2O should have been done with and without SCRV treatment, so that the reader can assess whether SCRV induces additional IFN activation (on top of MDA5 and STING autoactivation). I would recommend the authors include a sentence in the text to explain that ectopic expression of MDA5 or STING (i.e. overexpression from a plasmid) induces autoactivation of these proteins. Therefore, the IFN induction that is seen in panel 2N is likely due to MDA5/STING overexpression. SCRV treatment may further boost IFN induction, but this cannot be assessed without the 'mock' conditions. This information will help the readers to interpret Fig. 2N and 2O correctly.

---

## [Author Response]

The following is the authors’ response to the previous reviews.

**eLife assessment**
The authors present evidence suggesting that MDA5 can substitute as a sensor for triphosphate RNA in a species that naturally lacks RIG-I. The key findings are potentially important for our understanding of the evolution of innate immune responses. Compared to an earlier version of the paper, the strength of evidence has improved but it is still partially incomplete due to a few key missing experiments and controls.

We would like to thank the editorial team for their positive comments and constructive suggestions on improving our manuscript. We have made further improvements based on the valuable suggestions of the reviewers, and we are pleased to send you the revised manuscript now. After revising the manuscript and further supplementing with experiments, we think that our existing data can support our claims.

**Public Reviews:**

**Reviewer #1 (Public Review):**
This study offers valuable insights into host-virus interactions, emphasizing the adaptability of the immune system. Readers should recognize the significance of MDA5 in potentially replacing RIG-I and the adversarial strategy employed by 5'ppp-RNA SCRV in degrading MDA5 mediated by m6A modification in different species, further indicating that m6A is a conservational process in the antiviral immune response.However, caution is warranted in extrapolating these findings universally, given the dynamic nature of host-virus dynamics. The study provides a snapshot into the complexity of these interactions, but further research is needed to validate and extend these insights, considering potential variations across viral species and environmental contexts. Additionally, it is noted that the main claims put forth in the manuscript are only partially supported by the data presented.

After meticulous revisions of the manuscript, including adjustments to the title, abstract, results, and discussion, the main claim of our study now is the arm race between the MDA5 receptor and SCRV virus in a lower vertebrate fish, *M. miiuy*. This mainly includes two parts: Firstly, the MDA5 of *M. miiuy* can recognize virus invasion and initiate host immune response by recognizing the triphosphate structure of SCRV. Secondly, as an adversarial strategy, 5’ppp-RNA SCRV virus can utilize the m6A mechanism to degrade MDA5 in *M. miiuy*. Based on the reviewer's suggestions, we have further supplemented the critical experiments (Figure 3F-3G, Figure 4D, Figure 5G) and provided a more detailed and accurate explanation of the experimental conclusions, we believe that our existing manuscript can support our main claims. In addition, because virus-host coevolution complicates the derivation of universal conclusions, we will further expand our insights in future research.

**Reviewer #2 (Public Review):**
This manuscript by Geng et al. aims to demonstrate that MDA5 compensates for the loss of RIG-I in certain species, such as teleost fish miiuy croaker. The authors use siniperca cheats rhabdovirus (SCRV) and poly(I:C) to demonstrate that these RNA ligands induce an IFN response in an MDA5-dependent manner in m.miiuy derived cells. Furthermore, they show that MDA5 requires its RD domain to directly bind to SCRV RNA and to induce an IFN response. They use in vitro synthesized RNA with a 5'triphosphate (or lacking a 5'triphosphate as a control) to demonstrate that MDA5 can directly bind to 5'-triphosphorylated RNA. The second part of the paper is devoted to m6A modification of MDA5 transcripts by SCRV as an immune evasion strategy. The authors demonstrate that the modification of MDA5 with m6A is increased upon infection and that this causes increased decay of MDA5 and consequently a decreased IFN response.One critical caveat in this study is that it does not address whether ppp-SCRV RNA induces IRF3-dimerization and type I IFN induction in an MDA5 dependent manner. The data demonstrate that mmiMDA5 can bind to triphosphorylated RNA (Fig. 4D). In addition, triphosphorylated RNA can dimerize IRF3 (4C). However, a key experiment that ties these two observations together is missing.Specifically, although Fig. 4C demonstrates that 5'ppp-SCRV RNA induces dimerization (unlike its dephosphorylated or capped derivatives), this does not proof that this happens in an MDA5-dependent manner. This experiment should have been done in WT and siMDA5 MKC cells side-by-side to demonstrate that the IRF3 dimerization that is observed here is mediated by MDA5 and not by another (unknown) protein. The same holds true for Fig. 4J.

Thank you for the referee's professional suggestions. In fact, we have transfected SCRV RNA into WT and si-MDA5 MKC cells, and subsequently assessed the dimerization of IRF3 and the IFN response (Figure 2P-2Q). The results indicated that knockdown of MDA5 prevents immune activation of SCRV RNA. However, considering the potential for SCRV RNA to activate immunity independent of the triphosphate structure, this experimental observation does not comprehensively establish the MDA5-dependent induction of IRF3 dimer by 5’ppp-RNA. Accordingly, in accordance with the referee's recommendation, we proceeded to investigate the inducible activity of 5'ppp-SCRV on IRF3 dimerization in WT and si-MDA5 MKC cells, revealing that 5'ppp-SCRV indeed elicits immunity in an MDA5-dependent manner (Figure 4D). Additionally, poly(I:C)-HMW, a known ligand for MDA5, demonstrated a residual, albeit attenuated, activation of IRF3 following MDA5 knockdown, potentially attributed to its capacity to stimulate immunity through alternative pathways such as TLR3.

- Fig 1C-D: these experiments are not sufficiently convincing, i.e. the difference in IRF3 dimerization between VSV-RNA and VSV-RNA+CIAP transfection is minimal.

We have reconstituted the necessary materials and repeated the pertinent experiments depicted in Fig 1C-1D. The results demonstrate that SCRV-RNA+CIAP and VSV-RNA+CIAP exhibit a mitigating effect on the induction activity of SCRV-RNA and VSV-RNA on IRF3 dimerization, albeit without complete elimination (Figure 1C and 1D). These findings suggest the presence of receptors within *M. miiuy* and *G. gallus* capable of recognizing the viral triphosphate structure; however, it is worth noting that RNA derived from SCRV and VSV viruses does not exclusively depend on the triphosphate structure to activate the host's antiviral response.

Fig. 2N and 2O: why did the authors decide to use overexpression of MDA5 to assess the impact of STING on MDA5-mediated IFN induction? This should have been done in cells transfected with SCRV or polyIC (as in 2D-G) or in infected cells (as in 2H-K). In addition, it is a pity that the authors did not include an siMAVS condition alongside siSTING, to investigate the relative contribution of MAVS versus STING to the MDA5-mediated IFN response. Panel O suggests that the IFN response is completely dependent on STING, which is hard to envision.

In our previous laboratory investigations, we have substantiated the induction effect of STING on IFN under SCRV infection or poly(I:C) stimulation, as documented in the relevant literature (10.1007/s11427-020-1789-5), which we have referenced in our manuscript (lines 177-178). While we did assess the impact of STING on MDA5-mediated IFN induction in SCRV-infected cells, as indicated in the figure legends, we have revised Figure 2N-2O for improved clarity, and similarly, Figure 1H-1I has also been updated. Furthermore, considering that RNA virus infection can activate the cGAS/STING axis (10.3389/fcimb.2023.1172739) and the significant role of MAVS in sensing RNA virus invasion in the NLR pathway (10.1038/ni.1782), it is challenging to ascertain the respective contributions of STING and MAVS to the immune signaling cascade mediated by MDA5 during RNA virus infection. We intend to explore this aspect further in future research endeavors.

Fig. 3F and 3G: where are the mock-transfected/infected conditions? Given that ectopic expression of hMDA5 is known to cause autoactivation of the IFN pathway, the baseline ISG levels should be shown (ie. In absence of a stimulus or infection). Normalization of the data does not reveal whether this is the case and is therefore misleading.

Based on the reviewer's suggestions, we have rerun the experiment. We examined the effects of MDA5 and MDA5-ΔRD on antiviral factors in both uninfected, SCRV-infected, and poly(I:C)-HMW-stimulated MKC cells. Results showed that overexpression of both MDA5 and MDA5-ΔRD stimulated the expression of antiviral genes. However, when cells were infected or stimulated with SCRV or poly(I:C)-HMW, only the overexpression of MDA5, not MDA5-ΔRD, significantly increased the expression of antiviral genes (Figure 3F-3I).

Fig. 4F and 4G: can the authors please indicate in the figure which area of the gel is relevant here? The band that runs halfway the gel? If so, the effects described in the text are not supported by the data (i.e. the 5'OH-SCRV and 5'pppGG-SCRV appear to compete with Bio-5'ppp-SCRV as well as 5'ppp-SCRV).

Apologies for any confusion. The relevant areas in the gel pertaining to the experimental findings were denoted with asterisks and elaborated upon in the figure legends (Figure 4G, 4H, and 4M). The findings indicated that 5'ppp-SCRV, in contrast to 5'OH-SCRV and 5'pppGG-SCRV, demonstrated the ability to compete with bio-5'ppp-SCRV.

My concerns about Fig. 5 remain unaltered. The fact that MDA5 is an ISG explains its increased expression and increased methylation pattern. The authors should at the very least mention in their text that MDA5 is an ISG and that their observations may be partially explained by this fact.

First, as our m6A change analysis pipeline controls for changes in gene expression, these data should represent true changes in m6A modification rather than changes in the expression of m6A-modified transcripts (10.1038/s41598-020-63355-3). Similar studies demonstrated that m6A modification in RIOK3 and CIRBP mRNAs are altered following *Flaviviridae* infection (10.1016/j.molcel.2019.11.007). The specific calculation method is as follows: relative m6A level for each transcript was calculated as the percent of input in each condition normalized to that of the respective positive control spike-in. Fold change of enrichment was calculated with mock samples normalized to 1. Therefore, changes in the expression level of MDA5 can partially explain the increase in m6A modification on all MDA5 mRNA in cells, but it cannot indicate changes in m6A modification on each mDA5 transcript. We have supplemented the calculation method process in the manuscript and cited relevant literature (Lines 606-608). In addition, we have elaborated on the fact that MDA5 is an ISG gene in the experimental results (lines 260-261), and emphasized its compatibility with enhanced m6A modification of MDA5 in the discussion section (lines 405-409).

**Reviewer #3 (Public Review):**
In this manuscript, the authors explored the interaction between the pattern recognition receptor MDA5 and 5'ppp-RNA in the Miiuy croaker. They found that MDA5 can serve as a substitute for RIG-I in detecting 5'ppp-RNA of Siniperca cheilinus rhabdovirus (SCRV) when RIG-I is absent in Miiuy croaker. Furthermore, they observed MDA5's recognition of 5'ppp-RNA in chickens (Gallus gallus), a species lacking RIG-I. Additionally, the authors documented that MDA5's functionality can be compromised by m6A-mediated methylation and degradation of MDA5 mRNA, orchestrated by the METTL3/14-YTHDF2/3 regulatory network in Miiuy croaker during SCRV infection. This impairment compromises the innate antiviral immunity of fish, facilitating SCRV's immune evasion. These findings offer valuable insights into the adaptation and functional diversity of innate antiviral mechanisms in vertebrates.

We extend our sincere appreciation for your professional comments and insightful suggestions on our manuscript, as they have significantly contributed to enhancing its quality.

**Recommendations for the authors:**

**Reviewer #1 (Recommendations For The Authors):**
(1) The interpretation of Figures 1H and I, along with the captions, seems unclear. Particularly, understanding the meaning of the X-axis in Figure I is challenging. Additionally, the designation of "H2O = 1" on the Y-axis in Figure 1E lacks clarity. It would be helpful if the author could revise and clarify these figures for better comprehension.

We appreciate your reminder and have corrected and clarified these figures and figure legends (lines 768-772). We have replaced the Y-axis of Figure 1I with "Relative mRNA expression" instead of " Relative IFN-1 expression" (Figure 1I). In addition, we have added an explanation of "H2O=1" in the legend of Figure 1E.

(2) The interpretation of Figure 5 in section 2.5 seems incomplete. The author mentioned that both m6A levels and MDA5 expression levels are increased (lines 256-257), prompting questions about the relationship between m6A and MDA5 expression. If higher m6A levels typically lead to MDA5 mRNA instability and lower MDA5 expression, observing both increasing simultaneously appears contradictory. Considering the dynamic changes shown in Figure 5, it would be more appropriate to propose an alteration in both m6A levels and MDA5 expression levels. Given the fluctuating nature of these changes, definitively labeling them as solely "increased" is challenging. Therefore, offering a nuanced interpretation of the results and clarifying this aspect would bolster the study's conclusions.

While changes in m6A modification and the expression of m6A-modified transcripts are biologically relevant, identifying bona fide m6A alterations during viral infection will allow us to understand how m6A modification of cellular mRNA is regulated. As our m6A change analysis pipeline controls for changes in gene expression, these data should represent true changes in m6A modification rather than changes in the expression of m6A-modified transcripts (10.1038/s41598-020-63355-3). Similar studies demonstrated that m6A modification in RIOK3 and CIRBP mRNAs are altered following Flaviviridae infection (10.1016/j.molcel.2019.11.007). The specific calculation method is as follows: relative m6A level for each transcript was calculated as the percent of input in each condition normalized to that of the respective positive control spike-in. Fold change of enrichment was calculated with mock samples normalized to 1. Therefore, the upregulation of MDA5 expression can partially explain the increase in m6A modification on all MDA5 mRNA in cells, but it cannot indicate changes in m6A modification on each mDA5 transcript. We have supplemented the calculation method process in the manuscript and cited relevant literature. I hope to receive your understanding.

In addition, although higher m6A levels often lead to unstable MDA5 mRNA and lower MDA5 expression, SCRV can affect MDA5 expression through multiple pathways. For example, since MDA5 is an interferon-stimulated gene, the infection of SCRV virus can cause strong expression of interferon and indirectly induce high-level expression of MDA5. Therefore, the expression of MDA5 is not contradictory to the simultaneous increase in MDA5 modification (24 h). In order to further enhance our experimental conclusions, we supplemented the dual fluorescence experiment. The results indicate that, the infection of SCRV can inhibit the fluorescence activity of MDA5-exon1 reporter plasmids containing m6A sites but not including the promoter sequence of the MDA5 gene, and this inhibitory effect can be counteracted by cycloleucine (CL, an amino acid analogue that can inhibit m6A modification) (Figure 5G). This further indicates that SCRV can reduce the expression of MDA5 through the m6A pathway.

Finally, in light of the fluctuations in MDA5 expression levels, we have changed the subheadings of Results 2.5 section and provided a more comprehensive and precise elucidation of the experimental outcomes. We are grateful for your valuable feedback.

(3) In the discussion section, it would indeed be advantageous for the author to explore the novelty of this work more comprehensively, moving beyond merely acknowledging the widespread loss of RIG-I and suggesting MDA5 as a compensatory mechanism. Considering the well-established roles of MDA5 and m6A in host-virus interactions, the findings of this study may seem familiar in light of previous research. To enhance the discussion, it would be valuable for the author to delve into the implications of this evolutionary model. For instance, does the compensation or loss of RIG-I impact a species' susceptibility to specific types of viruses? Exploring such questions would provide insight into the broader significance of this compensation model and its potential effects on host-virus interactions, thus adding depth to the study's contribution.

We appreciate the expert advice provided by the referee. In response, we have expanded our discussion in the relevant section, addressing the potential influence of RIG-I deficiency and MDA5 compensation on the antiviral immune system in vertebrates (lines 371-376). Furthermore, we underscore the significance of exploring the impact of SCRV infection on MDA5 m6A modification, considering its compatibility with MDA5 as an ISG gene, in elucidating the host response to viral infection (lines 405-409).

(4) To improve the manuscript, it would be beneficial if the editors could aid the author in refining the language. Many descriptions in the article are overly redundant, and there should be appropriate differentiation between experimental methods and results.

We appreciate the reviewer’s comment. We have carefully revised the manuscript and removed redundant descriptions in the experimental results and methods.

**Reviewer #3 (Recommendations For The Authors):**
The authors have addressed all of my concerns.